# Sensory sharpening and semantic prediction errors unify competing models of predictive processing in human speech comprehension

Fabian Schneider [1,2]*, Helen Blank[1,2,3]

1 Institute for Systems Neuroscience, University Medical Centre Hamburg-Eppendorf, Hamburg, Germany, 2 Faculty of Psychology, Ruhr University Bochum, Bochum, Germany, 3 Research Center One Health Ruhr, University Alliance Ruhr, Bochum, Germany

* fabian@fschneider.dev

## Abstract

The human brain makes abundant predictions in speech comprehension that, in real-world conversations, depend on conversational partners. Yet, tested models of predictive processing diverge on how such predictions are integrated with incoming speech: The brain may emphasise either expected information through sharpening or unexpected information through prediction error. We reconcile these views through direct neural evidence from electroencephalography showing that both mechanisms operate at different hierarchical levels during speech perception. Across multiple experiments, participants heard identical ambiguous speech in different speaker contexts. Using speech decoding, we show that listeners learn speaker-specific semantic priors, which sharpen sensory representations by pulling them toward expected acoustic signals. In contrast, encoding models leveraging pretrained transformers reveal that prediction errors emerge at higher linguistic levels. These findings support a unified model of predictive processing, wherein sharpening and prediction errors coexist at distinct hierarchical levels to facilitate both robust perception and adaptive world models.

## 1 Introduction

The human brain continuously predicts incoming sensory inputs from generative models of the world, allowing for rapid and efficient perception [1,2]. In the domain of speech perception, predictive mechanisms have recently been demonstrated to operate across multiple levels, from low-level acoustic features to high-level semantic structure [3–5]. Yet, most studies have focused on speech perception in monologic contexts such as audiobooks, leaving open how predictive processes unfold in actual conversational settings, in which our main goal is to understand another person. In natural dialogue, predictions may track not only general linguistic statistics but also, crucially, leverage speaker-specific semantic priors, i.e., our expectations of what the other person is going to talk about. For example, if Jamie habitually talks about food,

**Data availability statement:** All code for and data from experiments, preprocessing, and analysis is available from https://doi.org/10.17605/OSF.IO/SNXQM.

**Funding:** This work was funded by the Emmy Noether program of the Deutsche Forschungsgemeinschaft (German Research Foundation; Grant No DFG BL 1736/1-1 awarded to H.B.; https://www.dfg.de/en/research-funding/funding-opportunities/programmes/individual/emmy-noether). We acknowledge financial support from the Open Access Publication Fund of UKE (Universitätsklinikum Hamburg-Eppendorf). Funders played no role in study design, data collection and analysis, decision to publish, or preparation of the manuscript.

**Competing interests:** The authors have declared that no competing interests exist.

speaker-specific semantic predictions may be crucial for correctly comprehending *Jamie: I'm putting my beans/dreams on the back-burner*, particularly under common challenging listening conditions such as noisy environments or poor phone lines [6]. However, whether such speaker-specific predictions occur remains untested.

While prominent theories like hierarchical predictive coding (hPC) propose that higher-level priors, such as semantic expectations, generate lower-level predictions, such as acoustic predictions, to facilitate perception and ambiguity resolution [1,7], how priors are applied at a mechanistic level is highly debated [2,8]: Empirically, prior work has reported enhancement of either expected components of the signal (sharpening), improving veridicality of representations [9–12], or unexpected parts of the signal (prediction error), improving informativeness of representations [13–16]. Yet, models of predictive processing typically assume distinct neuronal units for both sharpening and prediction error within regions [1,2]. Models tested in many empirical approaches, however, include only sharpening or prediction error components [9,11,12,15,16] and often do not track their dynamics jointly across time and levels of processing [10,13,14]. Therefore, it remains unclear how sharpening and prediction error computations are coordinated in function, over time, and across levels of the neural hierarchy [2,17]. Recent theoretical work predicts a hierarchical split: sharpening should dominate early sensory stages to stabilise perception, and prediction error should dominate later, higher-level stages to support adaptive updating of world models, but concrete evidence has been limited [18].

Here, we provide direct evidence that speaker-specific semantic priors modulate neural processing by sharpening early auditory representations, while generating prediction errors at the level of semantics. In a series of experiments, participants learned to identify acoustically ambiguous words in different speaker contexts that were cued with different faces. We show that participants apply speaker-specific semantic priors when resolving ambiguities, reporting to hear the word consistent with the speaker's semantic context. Using stimulus reconstruction models [19] and within-item composite representational similarity [20] regression of recorded electroencephalography (EEG) responses, we show that these priors manifest in early sharpening of neural representations, with low-level acoustic content shifted toward the expected acoustic signal. Further, single-trial encoding models [21,22] leveraging pretrained transformers [23] reveal that speaker-specific prediction errors emerge at the higher semantic level instead, albeit with a relatively early modulation of the EEG signal. Finally, we employ real-time modeling of semantic priors using free-energy models [24] to show that participants flexibly deploy and discard speaker-specific priors, depending on the probability of words under the prior. This dynamic interplay between flexible application of priors through low-level sharpening and high-level prediction errors provides new insights into how humans perceive and learn to adapt to individual speakers to maintain robust comprehension in acoustically ambiguous settings and critical evidence for a unified model of predictive processing in communication.

## 2 Results

To examine how listeners apply speaker-specific semantic priors, participants learned to associate six distinct faces with six unique semantic contexts, i.e., fashion, politics, arts, tech, nature, food, in two independent, preregistered experiments [25,26] (online $N_1 = 35$, EEG $N_2 = 35$, see Fig 1A). Each trial began with a speaker's face that cued a speaker-specific semantic context. Participants then heard a degraded morphed word that combined two words—only one of which fit the speaker's semantic context—and subsequently selected the perceived word from the two words making up the morph written on the screen (see Fig 1B).

As morphing two words may not result in perfectly ambiguous sounds, we measured context-free perception of each morph in an independent, preregistered experiment [27] ($N = 40$) and used these results as control predictors for acoustic unambiguities in subsequent analyses. In addition, we degraded the spoken words with a vocoding technique [28] so that the voice sounded like a whisper and voice-identity information was removed (see Stimulus creation). This allowed us to avoid any voice differences and link the identical spoken input to different faces, as we were interested in testing the effect of semantic priors on the representation of speech.

Speaker-specific feedback was provided visually by highlighting choices that were congruent or incongruent with the speaker in green or red, respectively, to ensure that each speaker was consistently linked to their semantic context (see Fig 1B). To track participants' semantic expectations throughout the experiment, we computed trial-by-trial estimates of their semantic priors by fitting Gaussian distributions over semantic embeddings of the words they reported to hear through free-energy models in real-time (see Estimating real-time semantic models). To be able to dissociate learning of speaker-specific semantics and general semantics across the experiment (e.g., learning what kinds of semantic contexts are included in the experiment across all speakers) or, to a lesser degree, sequence effects (e.g., responses at trial $t$ biasing responses at trial $t + 1$ irrespective of speaker-context), we fit two types of models: Firstly, modeling speaker-specific semantic priors, i.e. one prior per individual speaker, and, secondly, estimating speaker-invariant semantic priors, i.e., one general semantic prior across all six speaker contexts (see Fig 1D).

Critically, the same ambiguous morph (e.g., *sea-tea*) was presented in different speaker contexts (e.g., *nature-food*), allowing us to test whether perception and the underlying representation was pulled toward (i.e., sharpening) or away from the speaker priors (i.e., prediction error, see Fig 1D, 1E). For example, an idealised sharpened representation of the morph between *sea-tea* given a speaker who was associated with the semantic context *nature* would be more similar to *sea* and related words, whereas the idealised prediction error representation would be more similar to *tea* and related words instead.

### 2.1 Behavioural analysis shows that listeners apply speaker-specific semantic priors

In experiment one, participants ($N_1 = 35$, see Participants) were cued with a speaker's face and subsequently listened to a degraded spoken word (see Fig 1B). We modelled the probability of reporting to have heard any word as a function of trial number, the word's probability given the speaker and remaining acoustic biases of morphs. Participants reported to hear the word consistent with the speaker's semantic context (generalised linear mixed model (GLMM), $\beta = 1.64, s.e. = 0.12, z = 13.11, p = 2.71e^{-39}$, odds-ratio = 5.16), and this effect increased over trials ($\beta = 0.43, s.e. = 0.03, z = 12.77, p = 2.41e^{-37}$, odds-ratio = 1.54). Correspondingly, participants relied on acoustic unambiguities initially (GLMM, $\beta = -0.62, s.e. = 0.11, z = -5.81, p = 6.26e^{-9}$, odds-ratio = 0.54), but reliance on these decreased over trials (GLMM, $\beta = 0.39, s.e. = 0.06, z = 6.09, p = 1.14e^{-9}$, odds-ratio = 1.48; see S1 Table, S1 Fig).

In experiment two (different $N_2 = 35$, see Participants), we directly replicated these behavioural findings (see Fig 1C, S2 Table, S1 Fig) while also recording EEG responses. Together, these behavioural results demonstrate that listeners can learn and apply speaker-specific semantic priors to aid disambiguation in speech perception when speaker-context associations are reinforced with speaker-specific feedback (see Fig 1D). These behavioural findings cannot speak to the

PLOS Biology

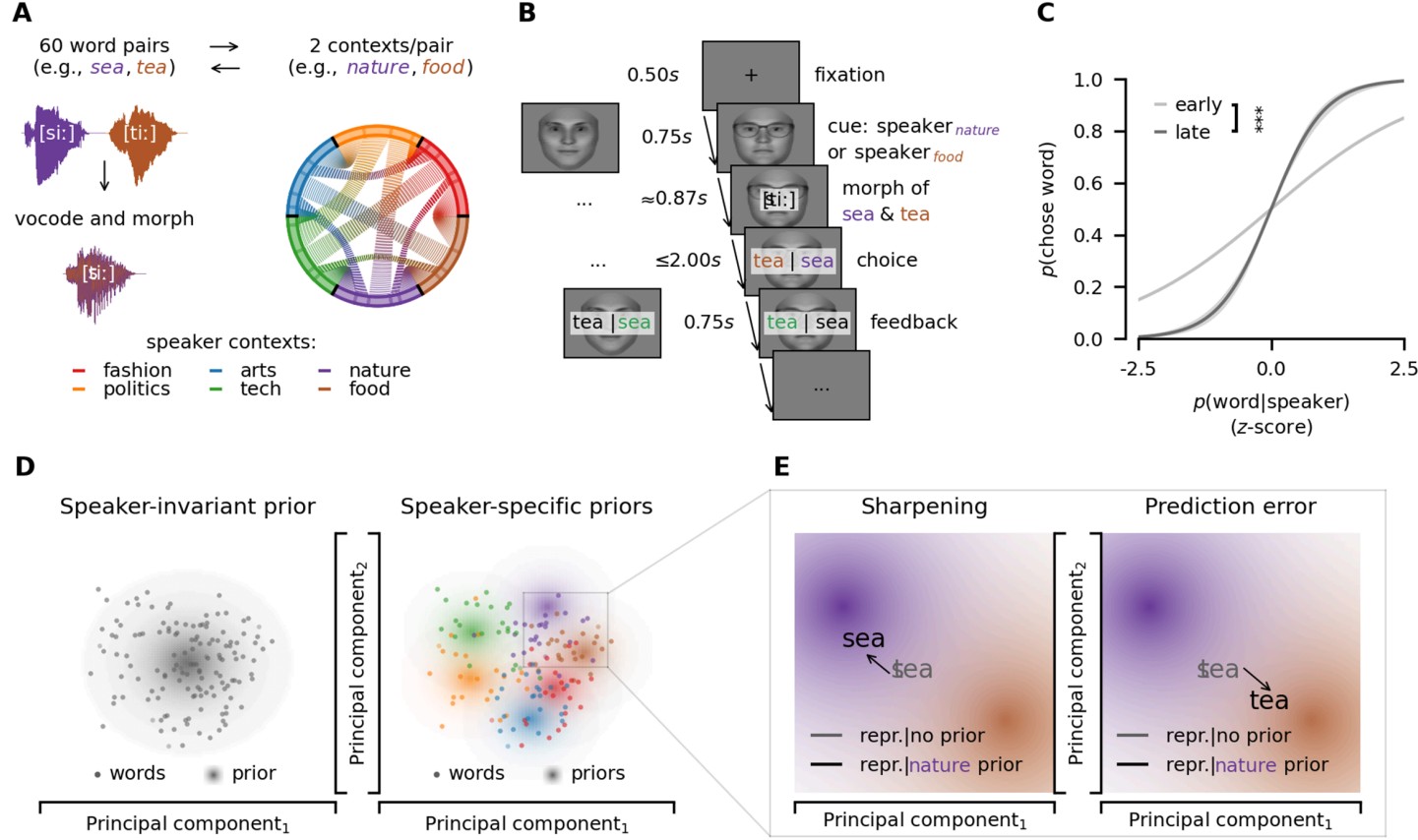

**Fig 1**. **Paradigm and behaviour. A** Stimuli. Sixty word pairs were created (e.g., *sea-tea*) where words within a pair sounded similar and could each be associated with one of six semantic contexts (e.g., *sea-tea* corresponding to *nature-food*). To decrease clarity and to induce ambiguity, words were slightly degraded to sound like a whisper and morphed into an intermediate acoustic signal between two words from two different contexts. Morphs were validated in a separate validation experiment (see Stimulus creation). **B** EEG part one with morphed words. Trials began with a fixation cross, followed by a visual speaker cue. Morphs were presented binaurally. Participants then indicated the word they had heard by button press. Finally, feedback was given in a speaker-specific manner such that each speaker could be associated with one specific semantic context. Faces were generated using FaceGen [29]. **C** Participants were more likely to report hearing a word when it was highly probable given the respective speaker (see Modeling choice behaviour). This preference increased over time. Lines are model predictions. Shaded areas around lines are 95%-confidence intervals. *** indicates $p \le 1e^{-3}$. **D** We tested two potential forms of priors (large dots) that might influence neural responses to words (small dots): First, participants may apply one general, i.e., speaker-invariant, prior reflecting general semantic expectations. Second, participants may learn and apply speaker-specific priors, i.e., six distinct priors, reflecting expectations about the individual speaker contexts. **E** In Bayesian models of the brain, there are two competing hypotheses about the application of priors: Firstly, prior expectations may pull neural representations towards the expected information (sharpening), i.e., when the speaker talking about nature is expected the representation of the morph (*sea-tea*) is shifted towards expected words (e.g., *sea*). Secondly, prior expectations may push neural representations towards more unexpected information (prediction error), i.e., when the speaker talking about nature is expected the representation of the morph (*sea-tea*) is shifted towards unexpected words (e.g., *tea*). Data and code supporting these findings are available from https://doi.org/10.17605/OSF.IO/SNXQM.

computational mechanisms at play, as speaker-specific feedback rendered consistent behavioural errors unlikely and neural mechanisms underlying representation may diverge from simple behavioural read-outs [17].

## 2.2 EEG analysis reveals that speaker-specific acoustic predictions sharpen sensory representations

Next we used EEG to investigate whether speaker-specific semantic priors modulate early sensory processing or high-level decision-making and whether they sharpen expected stimulus representations or induce prediction errors instead (see Fig 1D, 1E), potentially across different time points and stages of the processing hierarchy. To determine whether

speaker-specific semantic priors influence low-level sensory representations early into the neural processing hierarchy, we employed an approach combining stimulus reconstruction models and within-item composite representational similarity analysis (cRSA). In this combined approach, stimulus reconstruction models allowed us to isolate sensory representational content (see Stimulus reconstruction of morphs), and cRSA enabled a comparison of the geometry of this sensory content jointly with predictors in different representational formats such as acoustics and semantics (see Modeling acoustic expectations as top-k predictions; Regressing composite representational similarity).

First, gammatone filters, which model the human auditory system [30], were used to generate spectrograms reflecting the time-frequency representation of all morphed auditory stimuli. The high temporal resolution of EEG then enabled us to isolate sensory representations in the recorded neural data by correspondingly reconstructing gammatone spectrograms from EEG data following the presentation of the morphs (see Fig 2A, Stimulus reconstruction of morphs). We confirmed that spectrogram reconstruction from EEG was successful, as reconstruction performance, measured as the correlation between reconstructed spectrograms from the EEG and audio spectrograms from the corresponding morphs, was above chance-level overall (two-sided one-sample t-test, $M = 0.06, s.d. = 0.02, t(34) = 17.06, p = 8.96e^{-17}$; see Fig 2C) and in all individual frequency bands (two-sided one-sample t-test, all $p \leq 2.53e^{-6}$, see S2 Fig, S3 Table).

In a within-item cRSA approach, we computed cosine similarities of the reconstructed spectrograms from EEG signals following presentation of the identical morph in different contexts (e.g., *sea-tea|nature*, *sea-tea|food*, see Fig 2B) with the real audio spectrograms of the corresponding original words (e.g., *sea*, *tea*) to obtain a similarity matrix of the composite sensory representations (i.e., sensory cRSM).

We therefore measured within-item similarity, as opposed to between-item similarity as used in conventional RSA [20]. This approach was chosen because conventional RSA requires a commitment to specific mathematical formulations of the proposed computations for the construction of hypothesis RDMs [20]. However, theories of predictive processing allow for extremely high degrees of freedom in the precise formulation of their computations [7,8,31–33] which have been repeatedly criticised for hindering falsification [34–36]. Our experimental design, instead, aimed at differentiating between families of computations (see Fig 1A, 1B, 1D, 1E): Do sensory representations drift towards (i.e., common to sharpening computations) or away from predictions (i.e., common to prediction error computations)? Anchoring decoded representational content to real constituent acoustic spectrograms allowed us to directly test for directional shifts in representational content given the same auditory stimulus in different speaker contexts (see Regressing composite representational similarity) rather than testing geometric signatures of specific mathematical formulations in second-order comparisons used in conventional RSA [20]. While anchoring decoded representational content in real acoustic spectrograms may inflate absolute scores in cRSA regressions, relative explained variance between predictors on which our conclusions rely remains unbiassed.

To be able to assess the representational content in sensory cRSMs, we built two types of hypothesis cRSMs: Firstly, we built acoustic hypothesis cRSMs from spectrograms of clear words that were predictable from participants' speaker-specific and -invariant semantic priors. Specifically, we selected the five most probable words under a speaker prior and combined them into one probability-weighted audio spectrogram (for analysis of the influence of the number of predicted words, see S3 Fig, S4 Fig, S4 Table, S5 Table). This was done because predictions in the brain are inherently distributional [1]: The brain cannot truly foresee the target word at each trial. Given the potential acoustic heterogeneity even for semantically related words, predictions should reflect the distribution of expected acoustic features instead of the target word exclusively (see Modeling acoustic expectations as top-k predictions). Secondly, we generated semantic hypothesis cRSMs by comparing the semantic embeddings of the constituent words (e.g., *sea*, *tea* for morph *sea-tea*) with the current estimates of the speaker-specific and speaker-invariant semantic priors from free-energy models (see Fig 2B, see also Estimating real-time semantic models). Correlations between hypothesis cRSMs were confirmed to be low, ruling out significant variance inflation (see S5 Fig). We systematically regressed hypothesis cRSMs on sensory cRSMs in a hierarchical manner to assess the unique contributions of each hypothesis cRSM on sensory cRSMs, controlling for low-level

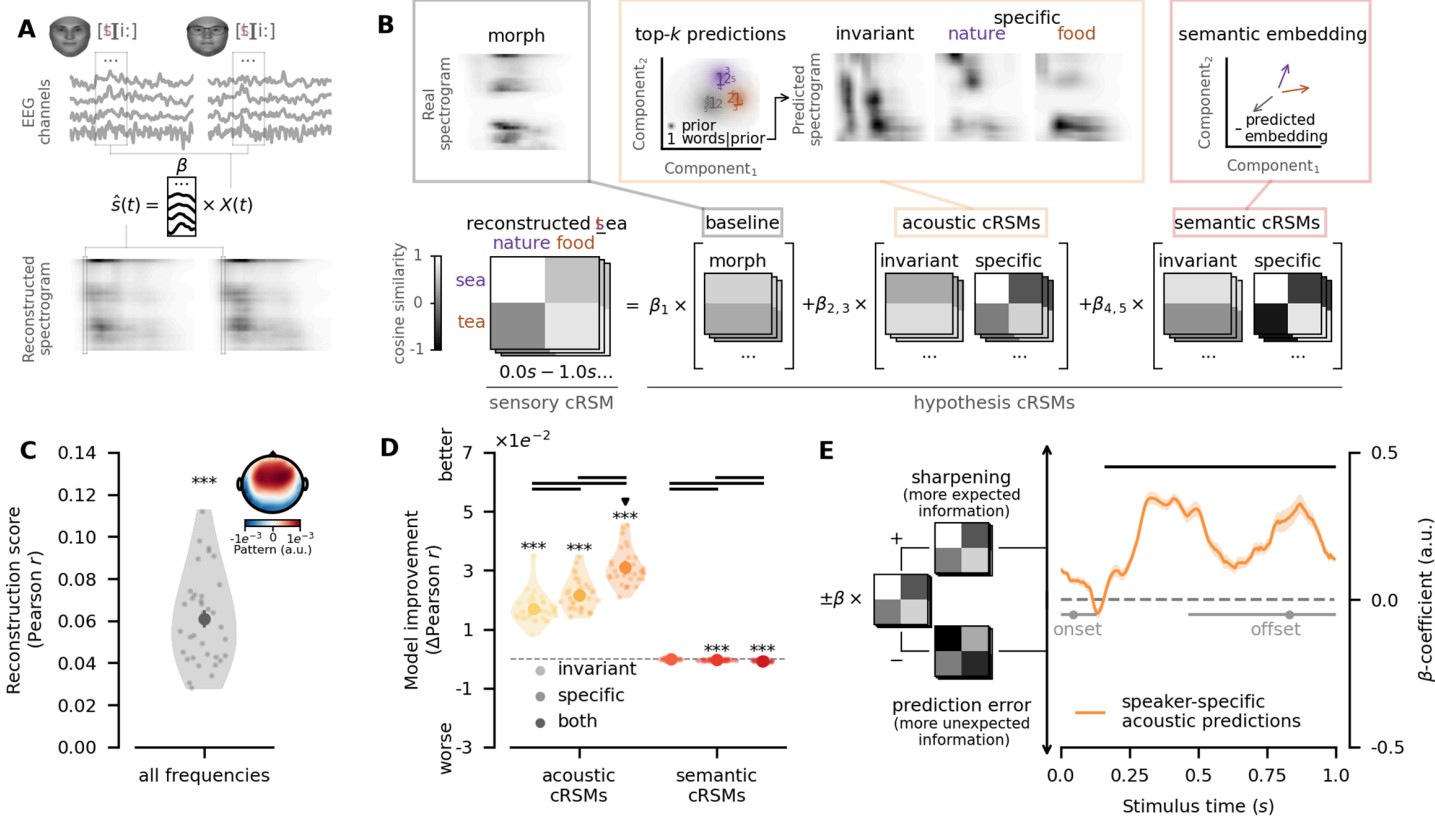

**Fig 2. Sensory sharpening at the acoustic level. A** Stimulus reconstruction models were used to decode acoustic signals $\hat{s}$ from neural activity $X$ during presentation of the same morph $s$ given both of its potential speaker contexts (e.g., *nature* and *food* for morph *sea-tea*, see Stimulus reconstruction of morphs). This allowed us to compare sensory content of neural representations directly in a within-item composite representational similarity analysis (see Regressing composite representational similarity). Faces were generated using FaceGen [29]. **B** Composite sensory representational similarity matrices (cRSMs) were computed from the reconstructions of the identical morph given two speaker contexts (e.g., *nature* and *food*) and the two spectrograms of the audio files of the original word pair. A set of hypothesis cRSMs were computed from the raw morph, top-5 acoustic predictions, and the current semantic predictions. Hypothesis cRSMs were systematically regressed on sensory cRSMs to test which combination of predictors best explained the similarity structure of decoded neural representations (see Regressing composite representational similarity). **C** Stimulus reconstruction models performed significantly above chance level. The inlay shows the average pattern that reconstruction models decoded in *z*-scored EEG signals. **D** cRSM regression revealed that both speaker-invariant and speaker-specific acoustic cRSMs improved the similarity structure of decoded neural representations, and that purely semantic cRSMs failed to do so. Improvements in out-of-sample prediction are visualised relative to baseline models. Circles indicate group means with error bars on circles indicating 95%-confidence intervals thereof. Transparent dots indicate single subjects. \*\*\*, \*\*, and \* indicate $p \leq 1e^{-3}$, $p \leq 1e^{-2}$ and $p \leq 5e^{-2}$, respectively. Note that significantly worse performance indicates evidence against a specific model, not in favour of its antithesis. Bold black lines between groups indicate $p \leq 5e^{-2}$. All *p*-values were corrected using the Bonferroni-Holm procedure. **E** Time-resolved coefficients of speaker-specific acoustic cRSMs in regressions showed consistently positive signs, indicating that neural representations were sharpened towards the more expected information present in hypothesis cRSMs. Lines represent means, with shaded areas around lines indicating 95%-confidence intervals. Bold black lines indicate $p \leq 5e^{-2}$. Grey dots represent the median on- and offsets of the first and last phonemes, respectively, with 95%-highest density intervals around them. Data and code supporting these findings are available from https://doi.org/10.17605/OSF.IO/SNXQM.

acoustic features and general linguistic statistics that were independent of the semantic priors manipulated in our experiment (see Fig 2B; see also Regressing composite representational similarity). We hypothesised that speaker-specific semantic priors generate predictions at the acoustic level, i.e., acoustic predictions, which either pull sensory representations towards the expected signal, i.e., make the representation more similar to the expected words (sharpening), or push them away from it, i.e., render the representation more similar to the unexpected words (prediction error, see Fig 1D, 1E).

cRSM regression revealed an effect of both speaker-specific and speaker-invariant semantic priors on acoustic representations. The sensory cRSMs were predicted more accurately when accounting for speaker-invariant and speaker-specific acoustic predictions (two-sided paired samples t-test, invariant-baseline: $t(34) = 22.40, p = 1.47e^{-20}$, specific-baseline: $t(34) = 25.12, p = 4.04e^{-22}$). Yet, speaker-specific acoustic predictions explained sensory cRSMs better than speaker-invariant acoustic predictions, indicating an increased reliance on speaker-specific information (two-sided paired samples t-test, specific-invariant: $t(34) = 4.18, p = 2.11e^{-3}$). Crucially, combining both types of acoustic cRSMs revealed their effects to be additive, with significantly better out-of-sample predictions than using either alone (two-sided paired samples t-test, both-baseline: $t(34) = 31.67, p = 2.33e^{-27}$, both-invariant: $t(34) = 17.87, p = 1.33e^{-17}$, both-specific: $t(34) = 15.03, p = 2.35e^{-15}$; see Fig 2D). While differences in correlation coefficients reported here were relatively small, differences were highly consistent across participants, and absolute differences were comparable to those commonly reported in RSA [17,20]. These findings suggest that predictions based on both speaker-invariant and speaker-specific semantic priors clearly shape the acoustic representations of heard words, though with a greater importance of speaker-specific priors.

To assess the specificity of these predictions to the acoustic level, we also tested whether purely semantic predictions would yield the same results. We entered the semantic embeddings used for top-5 predictions as semantic hypothesis cRSMs in the model, but omitted the step of generating specific acoustic predictions from these embeddings. Critically, purely semantic hypothesis cRSMs failed to improve out-of-sample predictions (two-sided paired samples t-test, invariant-baseline: $t(34) = -2.19, p = 0.25$, specific-baseline: $t(34) = -10.88, p = 1.66e^{-11}$, both-baseline: $t(34) = -7.93, p = 3.69e^{-8}$; see Fig 2D). These results indicate that speaker-specific semantic priors alter sensory representations at the acoustic level.

Next, we asked how speaker-specific predictions are combined with the incoming sensory signals at the acoustic level. We examined whether the neural representations reflected sharpening or prediction error by using the best regression model's $\beta$-coefficients, i.e., the model incorporating both speaker-invariant and -specific acoustic hypothesis cRSMs. Our hypothesis cRSMs were designed to reflect a pattern wherein more expected information is represented in positive values. In turn, due to the geometry of cosine similarities, this means that a negative sign inverts this pattern to represent more unexpected information (see Fig 2E). Consequently, positive and negative coefficients in our models naturally correspond to sharpening and prediction error computations, respectively, which allowed us to directly test the directionality of the effect of acoustic predictions on sensory representations. Cluster-based permutation tests revealed significant positive coefficients for speaker-specific acoustic predictions ($p \leq 8.00e^{-4}$) which corresponded to a cluster between $165ms$-$1000ms$ ($M = 0.22, s.d. = 0.04$; see Fig 2E). These findings demonstrate that speaker-specific semantic priors sharpen neural representations by pulling neural representations towards the expected acoustic signal.

We addressed several potential confounding factors: To rule out that our results depend on, firstly, the specific number of predicted words, we refit the best cRSM regression model while systematically varying the number of top-$k$ predictions from 1 to 19, as there were 20 words per speaker. This analysis revealed significant stepwise improvements in out-of-sample performance, with diminishing returns for larger $k$ (see S3 Fig, S4 Table). We verified that our results were robust by repeating all previous analyses using $k = 19$ predictions (see S4 Fig, S5 Table). Secondly, we confirmed that top-$k$ acoustic predictions captured meaningful acoustic expectations of participants that could not be explained equally well by top-$k$ predictions that were not participant-specific (see Modeling acoustic expectations as top-k predictions, Validating top-k acoustic predictions, S6 Fig, S6 Table). Further, we replicated our key finding of the dominance of acoustic sharpening at the sensory level in a conventional between-item RSA using $k = 1$ to rule out results being dependent on, thirdly, anchoring of hypothesis and sensory cRSMs to the same real spectrograms or, fourthly, averaging of spectrogram features (see Conventional representational similarity analysis, S7 Fig, S7 Table). Finally, we also verified results do not depend on the length of acoustic predictions (see Conventional representational similarity analysis, S8 Fig, S8 Table).

## 2.3 Speaker-specific semantic priors induce prediction error representations at higher levels

Building on our finding that speaker-specific semantic priors sharpen low-level sensory representations, we next examined whether prediction errors emerge at higher levels of the speech-processing hierarchy instead. To do this, we built encoding models (i.e., multivariate temporal response functions [19,21,22], mTRFs) of single-trial EEG signals that allowed us to probe signatures of acoustic and semantic prediction errors in the broadband EEG response. Critically, modeling broadband EEG responses rather than decoded sensory representations allowed us to probe whether prediction errors still co-occurred at, for example, the acoustic or semantic levels as abstract information theoretic signals (i.e., surprisal) disjoint from sensory representations. Here, mTRF models were chosen in lieu of more complex cRSM regression because information theoretic measures are known to be directly detectable and are commonly probed in single-trial EEG signals [37,38]. Since single-trial EEG signals are strongly influenced by stimulus properties such as acoustic envelopes and edges and general information theoretic measures like general phonotactic, lexical and semantic prediction errors, irrespective of the priors in our experiment [4,15], we implemented baseline encoding models to control for these general properties and language-specific prediction errors.

However, measuring prediction errors for morphs at levels higher than acoustics presents a unique challenge, as these stimuli are ill-defined at the phoneme and word level. To address this, we utilised recent advances in pretrained transformers [23], which allowed us to approximate general prediction error signals by feeding the raw audio of the morphs into these models and extracting 5-dimensional subspace projections of their layer activations [39,40]. Before feeding projected activations into encoding models, we verified that projected model activations were sensitive to general phonotactic, lexical and semantic prediction errors, and selected the best layer through back-to-back decoding [41]. This allowed us to disentangle the unique variance explained by each measure of prediction error in spite of existing correlations, in an independent auditory dataset (see Pretrained transformers as statistical surrogates, see also S9 Fig).

In line with prior research [4,15], we framed prediction errors in terms of surprisal, i.e., the negative log probability. We then tested whether single-trial EEG signals could be predicted from speaker-specific or -invariant acoustic or semantic surprisal beyond baseline models that included general stimulus properties and projected transformer activations that controlled for general surprisal, irrespective of the semantic priors in our experiment (see Fig 3A, Encoding of single-trial EEG data).

Results revealed that prediction errors emerged only at higher levels and only with respect to speaker-specific semantic priors. Concretely, speaker-invariant surprisal failed to improve encoding performance, irrespective of the hierarchical level (two-sided paired samples t-test, acoustic-baseline: $t(34) = -1.58, p = 0.49$, semantic-baseline: $t(34) = 1.11, p = 0.55$, both-baseline: $t(34) = 0.16, p = 0.87$). Crucially, speaker-specific surprisal improved encoding performance, but only at the semantic level (two-sided paired samples t-test, acoustic-baseline: $t(34) = -6.30, p = 1.78e^{-6}$, semantic-baseline: $t(34) = 5.33, p = 1.90e^{-5}$, both-baseline: $t(34) = 1.81, p = 0.08$, see Fig 3B). This suggests that while sharpening of speaker-specific representations occurs at the acoustic level, the brain computes prediction errors not at the acoustic level, but at the semantic level where predictions had been generated instead.

To probe the spatiotemporal dynamics of speaker-specific semantic surprisal, we systematically eliminated the influence of speaker-specific semantic surprisal from the full model at temporal lags $\tau = \{-0.1, ..., 0.8\}s$, and measured the change in encoding performance of the full model over the reduced model relative to total encoding improvement of the full model over the baseline model, which yields an estimate of the variance explained at any time point and channel (see Estimating spatiotemporal contributions of predictors). This knock-out procedure allowed us to directly test which temporal lags had a robust impact on encoding performance rather than having to rely on mTRF coefficients that may also reflect at least some degree of overfitting [42]. Cluster-based permutation tests revealed significant variance explained by speaker-specific semantic surprisal ($p \leq 1.80e^{-3}$) corresponding to a cluster spanning all sensors between $150ms$-$630ms$ (see Fig 3C). This spatiotemporal cluster aligns with the interpretation that speaker-specific prediction errors emerge primarily at levels higher than acoustics within the linguistic hierarchy, such as at the phonological and semantic levels.

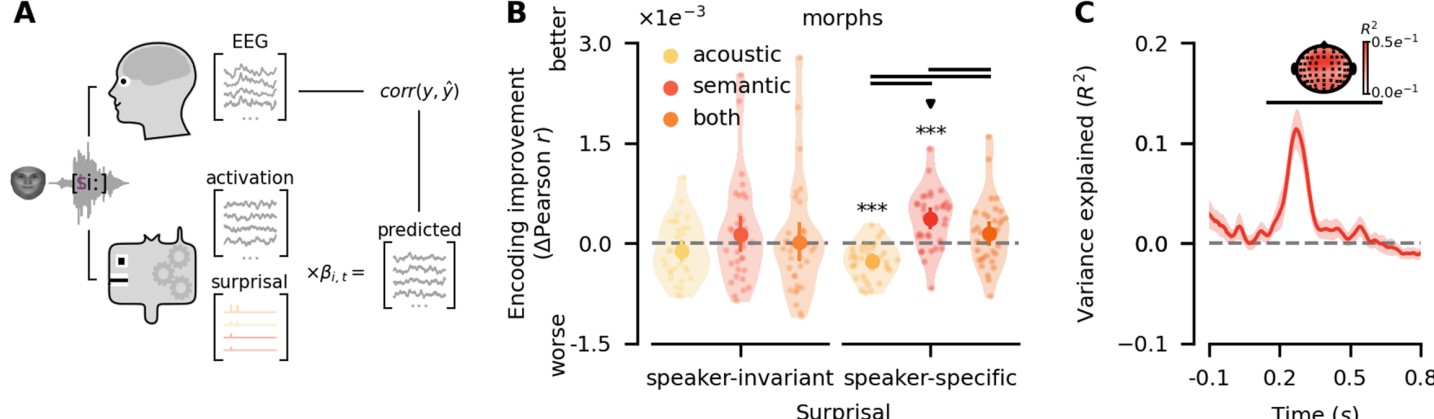

**Fig 3. Prediction errors at the semantic level. A** Pretrained transformers were employed to obtain control predictors for general phonotactic, lexical and semantic surprisal (see Pretrained transformers as statistical surrogates). Single-trial encoding of EEG responses was then applied to reconstruct observed EEG signals from control predictors and hypothesised speaker-invariant and speaker-specific surprisal predictors (see Encoding of single-trial EEG data). Faces were generated using FaceGen [29]. **B** Encoding models revealed that only speaker-specific semantic surprisal could significantly improve model performance. Encoding improvements are visualised relative to baseline models. Note that significant negative effects indicate that models failed to produce generalisable improvements. Circles indicate group means with error bars indicating 95%-confidence intervals thereof. Transparent dots represent subjects. ***, **, and * indicate $p \leq 1e^{-3}$, $p \leq 1^e - 2$ and $p \leq 5e^{-2}$, respectively. Bold black lines between groups indicate $p \leq 5e^{-2}$. All $p$-values were corrected using the Bonferroni-Holm procedure. **C** Spatiotemporally resolved contributions of speaker-specific semantic surprisal were estimated using a knock-out procedure (see Estimating spatiotemporal contributions of predictors). Speaker-specific semantic surprisal explained variance significantly with a corresponding cluster spanning a wide array of channels between $150ms$-$630ms$. Lines indicate the mean of variance explained, with shaded areas around them representing 95%-confidence intervals. Bold black lines indicate $p \leq 5e^{-2}$. The topography shows variance explained within each sensor, with channels contributing to the cluster highlighted in black. Data and code supporting these findings are available from https://doi.org/10.17605/OSF.IO/SNXQM.

Finally, we verified the robustness of these findings in two complementary analyses. First, increasing the dimensionality of the transformer activation subspace to 10 [39] to ensure that results were not an artefact of under-specified control predictors yielded consistent results (see S10 Fig, S9 Table). Second, we also verified that deriving control predictors for general properties and prediction errors from the target word rather than the morph's activation in the transformer produced consistent results to ensure that effects were not an artefact of poor performance of the transformer for morphed words (see S11 Fig, S10 Table). Together, these analyses confirm the reliability of our results and the emergence of speaker-specific prediction errors at the higher levels.

### 2.4 Flexible deployment of speaker-specific semantic priors

To test the flexibility of how speaker-specific semantic priors are applied during speech perception, we conducted an additional experimental task (EEG part two, see Fig 4A) in which we manipulated the degree to which the incoming sensory evidence conflicted with the preceding priors. Using real-time semantic models, we selected clear words where word$_1$ and word$_2$ were either highly congruent or highly incongruent with the speaker's expected semantics (e.g., *salad* vs. *senate* for *food*). Participants identified vocoded, but unmorphed, words following a speaker cue (see Fig 4A).

Participants were significantly slower in incongruent trials, suggesting increased surprisal or decisional conflict (linear mixed model (LMM), $\beta = 0.10, s.e. = 0.02, t(1606) = 5.41, p = 7.30e^{-8}$, see Fig 4B; see also Modeling behaviour in congruent and incongruent trials). Crucially, reaction times scaled with the probability of a word given the speaker in congruent trials (estimated marginal trends (EMT) at $\mathbb{E}[p(\text{word}|\text{speaker})], \beta = -0.12, s.e. = 0.02, t(9) = -5.53, p = 3.33e^{-4}$), but not in incongruent trials (EMT, $\beta = -0.02, s.e. = 0.02, t(14) = -0.89, p = 0.39$; see Fig 4B). For incongruent trials, this effect emerged only over trials (LMM, congruence $\times$ trial $\times$ $p$(word|speaker): $\beta = -0.03, s.e. = 0.02, t(3976) = -2.07, p = 3.83e^{-2}$).

PLOS Biology

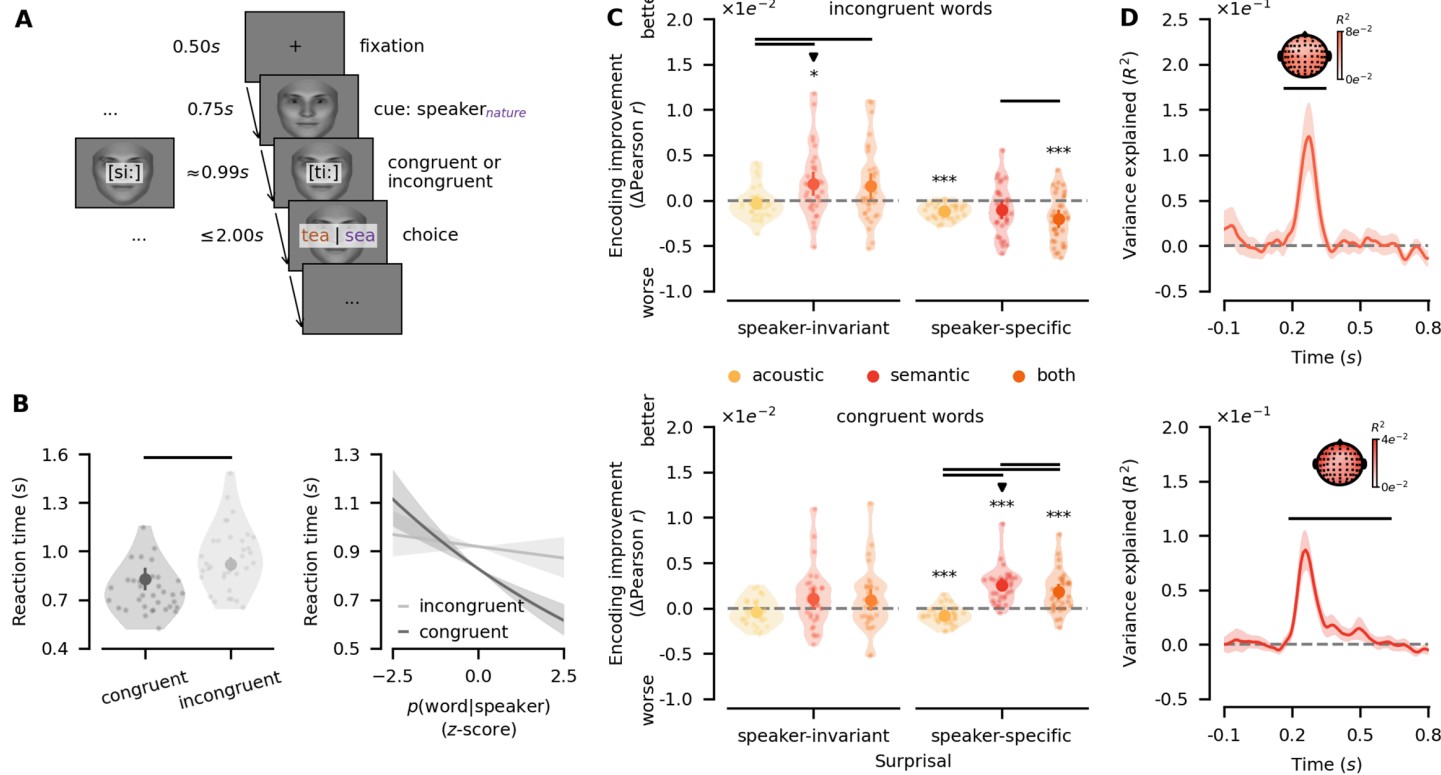

**Fig 4**. **Double dissociation between semantic congruency and prior specificity. A** EEG part two with congruent and incongruent words. A fixation cross was displayed, followed by the presentation of a visual speaker-cue. Highly congruent or incongruent vocoded words under the speaker prior were presented. Participants reported the word they heard. Faces were generated using FaceGen [29]. **B** Participants incurred a processing cost in incongruent trials. However, reaction times scaled with speaker-specific semantic surprisal only in congruent trials. Circles indicate group means with error bars indicating 95%-confidence intervals. Transparent dots indicate single subjects. Bold black lines between groups indicate $p \leq 5e^{-2}$. Shaded areas around lines indicate 95%-confidence intervals. **C** Encoding models revealed a double dissociation: In incongruent trials, only speaker-invariant semantic surprisal significantly improved encoding performance, whereas in congruent trials only speaker-specific semantic surprisal significantly boosted encoding performance. Encoding improvements are visualised relative to baseline models. Significant negative effects indicate that models failed to produce generalisable improvements. Circles indicate group means with error bars on circles showing 95%-confidence intervals. Transparent dots show individual participants. Bold black lines between groups indicate $p \leq 5e^{-2}$. ***, **, and * represent $p \leq 1e^{-3}$, $p \leq 1e^{-2}$ and $p \leq 5e^{-2}$, respectively. The downward-facing triangles indicate the best models by congruence. All p-values were corrected using the Bonferroni-Holm procedure. **D** Knock-out analysis of coefficients revealed significant variance explained by speaker-invariant semantic surprisal in incongruent trials (top) as well as by speaker-specific semantic surprisal in congruent trials (bottom). Speaker-invariant semantic surprisal was associated with a cluster across all channels between 170ms-345ms, and speaker-specific semantic surprisal was associated with a cluster across all channels between 190ms-635ms. Lines indicate means with shaded areas representing 95%-confidence intervals. Inlaid topographies show variance explained across sensors, with sensors contributing to clusters marked in black. Bold black lines indicate $p \leq 5e^{-2}$. Data and code supporting these findings are available from https://doi.org/10.17605/OSF.IO/SNXQM.

Thus, while participants continued to leverage speaker-specific priors, they seemed to discard them initially when encountering highly improbable words, incurring a switch cost.

EEG encoding models mirrored this pattern: In incongruent trials, only speaker-invariant semantic surprisal boosted encoding performance (semantic-baseline: $t(34) = 3.03, p = 1.88e^{-2}$; see Fig 4C), whereas in congruent trials only speaker-specific semantic surprisal significantly improved encoding performance (semantic-baseline: $t(34) = 8.39, p = 3.39e^{-8}$; see Fig 4C, see also S11 Table). Knock-out analyses revealed significant variance explained by speaker-invariant semantic surprisal in incongruent trials ($p \leq 1.40e^{-3}$), corresponding to a cluster across all channels between 170ms-345ms, and significant variance explained by speaker-specific semantic surprisal in congruent trials ($p \leq 1.40e^{-3}$),

corresponding to a cluster across all channels between 190*ms*-635*ms* (see Fig 4D). This double dissociation of semantic congruency and specificity of the semantic priors demonstrates that listeners dynamically deploy and discard speaker-specific priors depending on contextual plausibility.

## 3 Discussion

In this study, we investigated whether listeners utilise speaker-specific semantic priors to facilitate speech comprehension, and how such priors influence processing at different levels of the processing hierarchy. Specifically, we aimed to resolve a long-standing debate on whether predictive processing operates primarily via prediction error or sharpening mechanisms [1,10,13,17,18]. Our findings provide clear evidence for an integrated account in which both mechanisms coexist at distinct levels of the hierarchy, as suggested by more recent accounts of the Bayesian brain [17,18].

First, we demonstrate that listeners can apply speaker-specific semantic priors to disambiguate acoustically ambiguous words when speaker-context associations are consistently reinforced through feedback (see Fig 1C). Second, we show that these priors sharpen low-level sensory representations, actively shifting them towards the expected acoustic signal (see Fig 1E, Fig 2D, 2E). Third, while sharpening occurs at the acoustic level, prediction errors emerge at higher linguistic levels, such as semantics (see Fig 3B, 3C, Fig 4C, 4D). Finally, we find that speaker-specific semantic priors are not applied indiscriminately: when the likelihood of a given input under a speaker prior is exceedingly low, listeners flexibly discard the prior rather than generating extreme prediction errors and, consequently, potentially deleterious model updates (see Fig 4C, 4D).

These findings align with recent refinements of Bayesian models of the brain, which propose that sharpening occurs at lower levels while prediction errors dominate higher levels of the neural hierarchy [2,18]. This nuanced view departs from traditional models of hPC, which posit that only prediction errors should be propagated within the hierarchy [1,7]. Instead, our results support a synergistic account, wherein sharpening enhances perceptual robustness while prediction errors may enable adaptive updating of internal models [18].

This interpretation is supported by prior work in speech perception demonstrating sharpening to occur at the sensory level [9,12], though here explicit source modeling is lacking, and prediction errors in regions associated with higher-level processing such as phonology or semantics [13–15,38,43], though prediction errors were also identified in regions typically associated with acoustics [44]. Analogously, prior work in vision supports sharpening in sensory regions such as V1 [10,45], but there are conflicting reports about prediction errors through-out the processing hierarchy [17,46,47]. More recent work in vision suggests that both sharpening and prediction errors may occur in deep and superficial laminae of V1, respectively [48,49], which may partly explain conflicting prior findings and complement the view that sharpening and prediction errors co-occur in cortical layers but dominate at different levels of the neural hierarchy [17,18]. However, laminar differences remain untested in speech comprehension, and cannot fully explain discrepancies between dominance of sharpening or prediction errors without assuming some kind of gain modulation thereof [50].

Here, we report empirical support for sharpening at the acoustic level (see Fig 2E). Notably, the temporal extent of sharpening suggests that sharpening is not constrained to times of high uncertainty, for example, only for the disambiguating phonemes of a morph such as /t/ or /s/ for morph *tea-sea*, but occurs in parallel with the temporally unfolding acoustic signal instead. Sharpening at the acoustic level stands in contrast with previous findings emphasising the role of prediction error in speech processing [3,4,13–15]. This discrepancy may stem at least partly from methodological differences: whereas previous studies primarily analysed broadband EEG or BOLD responses, we employed speech decoding models [19] to isolate changes in low-level sensory representations. Supporting this view, evidence of sharpening in monologic contexts has previously been reported using similar sensory decoding approaches [9,12], albeit only for speech envelopes rather than low-level acoustic representations as captured with full spectrograms.

Our task was carefully constructed to test facilitation of speech comprehension through speaker-specific semantic priors that were reinforced through explicit speaker-specific feedback. Therefore, it is also conceivable that other tasks

demanding attention towards unexpected features, such as deviation detection, might shift low-level representations towards prediction error representations [50]. In complementary RSA analyses over decoded sensory representations, we also found some evidence of possible co-occurring prediction errors at the acoustic level that were, however, considerably weaker than sharpening responses (see S7 Fig, S8 Fig) and not apparent in the broadband EEG signal (see Figs 3–4). This would be in line with the view that sharpening and prediction error computations co-occur in sensory regions at different laminae [48,49] where we might expect task- or context-based gain modulation [50] to dictate the dominance of sharpening or prediction error at the sensory level. We speculate that, even in this view, sharpening should dominate in natural, dialogic settings, in line with a synergistic account of sharpening and prediction errors [18].

Yet, while these supplementary RSA results hint at weakly co-occurring acoustic prediction error at the sensory level (see S7 Fig, S8 Fig), we do not find evidence thereof in regressions of sensory cRSMs (see Fig 2D) or encoding models of broadband EEG responses (see Fig 3). In the case of cRSM regression, this is expected given that supplementary RSA results show acoustic sharpening to be considerably stronger than acoustic prediction errors (see S7 Fig): When investigating the directionality of the representational shift overall, the stronger sharpening effect should dominate the weaker prediction error effect. That encoding models of broadband EEG responses do not support acoustic prediction errors (see Fig 3D; Fig 4) demonstrates the critical role that analysis granularity may play. As with sharpening, it is possible that multivariate sensory prediction errors are not easily detectable in commonly employed broadband encoding of univariate distance metrics [15,38]. In all, investigating predictive processing in natural speech by incorporating a wider array of methodologies, particularly, placing greater importance on representational content, laminar resolution of activity, and task-demands, may allow comparing and disentangling sharpening and prediction error computations at the acoustic level across monologic and dialogic contexts more thoroughly. Within regions, sharpening and prediction errors may be represented in deep and superficial layers, respectively [48,49], and across regions sharpening may dominate prediction errors at the sensory level  [18] unless task-demands place greater importance on deviation detection [50].

Interestingly, here we find that speaker-specific prediction errors emerge relatively early, with clusters around $150ms$-$630ms$ and $190ms$-$635ms$ after stimulus onset in the main task testing perception of morphed, i.e., ambiguous words, and the follow-up task testing congruent and incongruent words, respectively (see Fig 3C, Fig 4D). Differences in temporal extent of these clusters are difficult to interpret, given that cluster-based permutation testing does not guarantee significance at specific time points [51,52]. However, peak latencies showed no differences between tasks, with peaks consistently around $252ms$ post stimulus onset (see S12 Fig). This is striking given that semantic prediction errors are typically thought to peak around $400ms$ post onset [53,54]. While semantic processing has previously been reported around 250ms [55,56], an alternative interpretation may be that prediction errors induced by semantic priors arise at multiple intermediate and higher-level stages, first at the phonemic and then the semantic levels, which would be in agreement with the prolonged time course of speaker-specific prediction errors. While this would also be consistent with hPC where prediction errors are propagated along the processing hierarchy [1,18], there is a long-standing debate whether EEG components around this time can even be separated into functionally distinct phonemic and semantic components [53,57]. Given the limited size of the stimulus set in this study, we cannot derive appropriate estimates of speaker-specific phonemic surprisal, leaving this question open for future research. Crucially, this does not change the interpretation of our findings with respect to hierarchical processing in models of the Bayesian brain.

With respect to both sharpening and prediction error computations reported here, two further avenues are of interest that fall beyond the scope of this paper: Firstly, source estimates would allow identification of their neural loci which we would expect to peak in A1, reflecting early auditory processing, and more distributed frontal and temporoparietal sources, reflecting later stages of semantic processing, respectively [53,54,58]. Secondly, theories of predictive coding have hypothesised distinct oscillatory signatures of predictions in lower and prediction errors in higher frequency bands [59,60] that may be investigated using MEG with individualised head casts to improve source estimates [61,62].

Our findings also highlight a key theoretical gap in predictive models: while it is commonly assumed that the brain constructs multiple types of priors at all levels of the processing hierarchy [1,7,18], the precise mechanism by which competing priors are selected remains unclear. The coexistence of priors at the same level, such as speaker-specific and speaker-invariant semantic priors, may be crucial for tracking multiple relevant statistical regularities, yet our results suggest that not all priors are applied simultaneously. Instead, we find a double dissociation of prior congruency and specificity at higher levels of the neural hierarchy, where priors are selected dynamically based on their relative likelihood. Specifically, a speaker-specific prior was only applied when the incoming word was congruent with that prior. This has important implications for predictive coding models, particularly in the context of world model updates [1,7]: Firstly, allowing, for example, detection of a new semantic topic when listening to the identical speaker. Secondly, if all high-level priors were applied and updated indiscriminately, they would risk collapsing into a single, undifferentiated representation. Thus, priors must be selectively updated, potentially through precision-weighting mechanisms that govern their influence.

Most previous studies on predictive processing have primarily examined speech perception in monologic contexts such as audiobooks [3,4,15], leaving open the question of how semantic predictions may operate in speaker-specific conversational settings. Our study investigated how predictions are shaped not only by general linguistic statistics but also by speaker-specific characteristics. Thereby, our approach extends and links to previous experimental findings showing that listeners adapt to individual speakers at, for example, the phonetic [63–65] and lexical levels [66–68]. Here we show for the first time that listeners also adapt to semantic patterns of individual speakers. As such, our findings also advance our understanding of how listeners may deal with common sources of uncertainty in the speech signal such as loud environments or poor phone lines [6].

Beyond theoretical implications, our findings have direct relevance for clinical applications, particularly in neural speech decoding and brain-computer interfaces (BCIs). Decoding approaches hold promise of restoring communicative abilities for patients with conditions such as aphasia, anarthria, or locked-in syndrome [69]. While recent advances have improved speech decoding accuracy [70–73], current methods remain data-intensive and lack the precision needed for seamless communication [72,73]. Our results suggest that incorporating individualised speaker-specific models could enhance decoding accuracy, particularly in interactive settings with clinicians or family members, where strong priors may shape communication. Furthermore, an intriguing question arising from our study is whether speaker-specific semantic priors also influence speech production. If listeners adjust their own speech to match their semantic expectations of a conversational partner, this adaptation could be leveraged in BCIs to improve bidirectional communication and hence may have significant implications for speech rehabilitation and augmentative communication technologies.

In conclusion, we show that listeners learn and apply speaker-specific semantic priors, which actively shape their perceptual experience and sharpen low-level sensory representations towards the expected acoustic signals. Prediction errors, in contrast, emerge at higher levels of the linguistic hierarchy, supporting a complementary role of sharpening and prediction errors in predictive processing. These findings refine our understanding of predictive processing in the brain and suggest new directions for clinical applications, as in the development of dialogic brain-computer interfaces.

## 4 Methods

The series of experiments of this study was preregistered [25–27]. Across an online and subsequent EEG experiment, participants learned to associate six distinct speakers with six unique semantic contexts (see Fig 1A). Participants heard identical degraded and morphed auditory stimuli given different speaker contexts (see Fig 1B), allowing us to probe the influence of speaker-specific semantic priors on neural representations along the speech processing hierarchy (see Fig 1D, 1E). In the EEG experiment, participants completed a subsequent follow-up task including degraded unmorphed words that were either particularly congruent or incongruent with their perceived semantic model of any given speaker, which allowed us to test the limits of prior application (see Fig 4A).

## 4.1 Participants

In the validation experiment, 40 volunteers were recruited through Prolific and completed the experiment online (20 female, age range 19-40, $M = 28.35 \pm 6.27$). Participants that failed to pass an initial headphone test [74] or failed to respond on ten of the preceding twenty trials were excluded in real-time. A priori power analysis suggested that this would yield sufficient power for $\chi^2$-tests to detect deviations by at least 0.15 from chance-level in individual morphs [27].

In experiment one, 35 different volunteers were recruited through Prolific and participated in the experiment online (17 female, age range 22-39, $M = 29.46 \pm 4.78$). Again, participants were excluded in real-time based on headphone and attention tests. Participants received £9/hr in compensation for their time. Sample size was determined through power simulations using PyMC5 [75] and ArviZ [76] based on previous experimental data that indicated sufficient power to detect behavioural effects bigger than log odds 0.15 [25].

In experiment two, 36 different volunteers completed the study in the EEG laboratory. One participant was excluded from the analysis due to excessive muscle-related artifacts. The final sample included 35 participants (20 female, age range 20-39, $M = 28.09 \pm 5.29$). Participants were compensated with 13€/hr. Sample size was determined through power simulations based on pilot recordings ($N = 2$) using PyTorch [77] that indicated sufficient power to detect moderate effect sizes in the neural data [26].

In all experiments, all participants were right-handed, native speakers of German, and reported no history of neurological disorders, ADHD, ASD, dyslexia, impaired hearing, or face or colour blindness. The study followed the Declaration of Helsinki, all experimental procedures were approved by the Ethics Committee of the Chamber of Physicians in Hamburg (approval no. PV7210), and participants provided written informed consent.

## 4.2 Stimulus creation

An initial list of 210 target words was created by searching the neighbourhood of six semantic contexts (food, fashion, technology, politics, arts, and nature) seeded at the embedding of the context's label in a GloVe embedding pretrained on German Wikipedia articles [78,79]. For each target word, one alternative word corresponding to one of the other five semantic contexts was chosen. Targets and alternatives were chosen such that phonemic differences would be minimal, although not all formed minimal pairs (e.g., sea/tea in food/nature). We purposefully selected pairs of words with minimal acoustic differences, each of which fit into one of six semantic contexts. This allowed us to balance the stimuli symmetrically: Either member of a word pair was used as the target while the other served as the alternative, depending on the preceding context (e.g., target for *sea-tea|nature*: *sea*, but target for *sea-tea|food*: *tea*).

A professional female speaker produced two enunciations of all words (duration: $M = 970ms \pm 144ms$). Audio files were pitch-corrected to roughly 170Hz to minimise cues about gender using Praat [80] and sound pressure levels were normalised to $-20LUFS$ to control for loudness. Next, we used a combination of vocoding and morphing the words twice to obtain two optimal 50%-morphs per pair, each with different noise profiles. Specifically, audio files were vocoded using twelve bands of white noise to remove most of the remaining voice properties from the signal [28]. This level of noise-vocoding was chosen because it has previously been shown to be intelligible, particularly given prior knowledge [13]. From the recorded utterances, we generated four 50%-morphs using TandemSTRAIGHT [81] after application of dynamic time warping [82,83] over constant-Q transformed [84] audio data that aligned targets and alternatives in time. We computed cosine similarities between morphs and clear words and selected the least biased morph per pair.

Within each pair, perception of a morph as one word or the other was controlled in a preregistered validation experiment [27]. Participants (see Participants) listened to all morphs and had to indicate which of the two visually presented options they had heard by button press. In total, participants completed 468 trials (6 contexts × 2 practice items + 2 repetitions × 120 morphs +2 repetitions × 6 contexts × 3 control items). Per morph, we computed a descriptive statistic $\kappa$ that captured the remaining perceptual bias of participants to perceive one word or the other within the pair underlying each

morph:

$$\kappa = \frac{p(\text{alternative}|\text{morph}) - \mu}{1 - \mu}$$

Note that $\mu$ represents chance level 0.50 here. We then dropped all morphs that did not meet $0.15 \leq \kappa \leq 0.85$ to remove biassed morphs that did not lead to ambiguous perception.

Next, for each target and alternative, the best semantic fits among the six candidate contexts were determined to find the subset of word pairs that maximised the coherence of each semantic context. To do this, we computed cosine similarities between all targets and all alternatives from all contexts (e.g., sea-tea= 0.12, sea-wattage= −0.06, sea-boat= 0.31, ...) and averaged them by context (e.g., sea-nature= 0.38, sea-food= 0.14, sea-fashion= −0.08, ...). We then ranked each pair by the semantic fit to either potential contexts $R_t, R_a$ and computed geometric means across ranks as $(R_t R_a)^{\frac{1}{2}}$ for each pair. We selected word pairs to minimise the geometric means, yielding a total of 60 pairs where $word_1$ was coherent with one of the six contexts and $word_2$ was coherent with another (see Fig 1). As a result of this procedure, we obtained twenty words per context ($6 \times 20 = 120$ words overall), each of which was morphed with one other word, yielding sixty morphs with two different noise profiles that showed no systematic perceptual biases (duration: $M = 867ms \pm 144ms$; for an overview, see S13 Fig; S12 Table). For all, i.e., unmorphed and morphed, words, we generated annotations of phoneme on- and offsets using MAUS [85].

Finally, a set of six face images was created using FaceGen [29] to obtain one visual cue per speaker-specific semantic context. Speaker images were crossed with six distinctive features (two types of glasses, scars, piercings each) to ease recognition and images were luminance-corrected.

### 4.3 Procedure

Stimuli were presented using jsPsych [86] in online experiments, whereas PsychoPy [87] was used in the EEG laboratory. Before all experiments, participants were introduced to the six speakers, each represented by a face and a one-sentence summary of their interests. The online experiment lasted ~45 minutes, whereas the EEG experiment lasted ~2.5 hours, including briefing, setup, passive listening (~20 minutes, not reported here), EEG part one with morphed words (~35 minutes), EEG part two with congruent and incongruent words (~15 minutes) and debriefing.

All experiments followed the same trial structure: Each trial began with a fixation cross ($500ms$), followed by the visual presentation of a speaker ($750ms$). Next, a vocoded morph was played ($M = 867ms \pm 144ms$ and, in EEG, $400ms$ delay) in the online experiment and EEG part one, whereas a vocoded congruent or incongruent word was played in EEG part two ($M = 990ms \pm147ms$ and $400ms$ delay). Participants were subsequently shown two word options, one congruent with the speaker's semantic context and one from a different context, and selected the word they perceived via button press (response window: $3000ms$ online, $2000ms$ in EEG). To reinforce speaker-specific semantic associations, participants received immediate speaker-specific feedback: their selection was highlighted in green if correct under the speaker and red if incorrect under the speaker ($750ms$, see Fig 1B). Feedback was omitted in the EEG part two. The inter-trial interval was randomly jittered between $1.0s$–$1.4s$. In the online experiment and EEG part one, participants completed a total of 270 trials (6 practice trials, 24 control trials, 2 repetitions × 2 contexts × 60 morphs main trials), with optional breaks every 20 trials. In EEG part two, participants completed 120 trials (60 congruent, 60 incongruent).

During the EEG experiment, real-time estimates of each participant's semantic priors were computed using a free-energy approach that were leveraged for sampling of congruent and incongruent stimuli in part two (see Estimating real-time semantic models). After completion of task one, individual semantic priors were used to sample subject- and speaker-specific word pairs. Each pair contained one word strongly congruent with a speaker's semantic context (i.e., $context_a$) and another word strongly incongruent with a different context (i.e., $context_b$). To maximize the difference in

semantic fit, word pairs were selected based on the following criterion:

$$\arg\max_{\text{word, context}} p(\text{word}_1|\text{context}_a) - p(\text{word}_2|\text{context}_b)$$

This ensured that $\text{word}_1$ was highly congruent with $\text{context}_a$, whereas $\text{word}_2$ was particularly incongruent with $\text{context}_b$. The sampling procedure also ensured that each context contained ten congruent-incongruent word pairs. This modification made the stimuli clearly identifiable as either congruent or incongruent, allowing us to test how speaker-specific priors are applied when word probability is either very high or very low for a given speaker.

### 4.4 EEG acquisition

EEG data were acquired using a 64-channel Ag/AgCl active electrode system (ActiCap64; Brain Products) that was placed according to the extended 10-20 system [88]. A total of sixty electrodes were placed centrally, with reference and ground electrodes at FCz and Iz, respectively. Four additional electrodes were placed to record the vertical and horizontal electrooculogram. Recordings were made with impedances $\leq 10k\Omega$. Data were sampled at $1000Hz$.

### 4.5 Estimating real-time semantic models

To dynamically track how participants formed and updated speaker-specific semantic priors during the experiment, we employed a real-time modeling approach based on free-energy optimisation [24]. This was done because the low computational cost allowed us to continuously estimate trial-by-trial shifts in participants' inferred semantic representations of speaker-specific and speaker-invariant priors $\pi$ for any given speaker $i$ in real-time. To do this, we sought to infer $\pi$ by maximising $p(\pi|S)$ for some semantic input $S$.

For efficiency in model estimation, we reduced the original 300-dimensional GloVe embeddings [78,79] to a 50-dimensional subspace that spanned all $S$. Briefly, we preprocessed demeaned embeddings by applying principal component analysis (PCA), and removing the top $D = 7$ components, and projected data to 50 dimensions, followed by a second iteration of the PCA-based preprocessing. This approach has been shown to improve embedding quality [89,90].

For simplicity, we assumed that $\pi$ follows a 50-dimensional Gaussian distribution per speaker, with mean $\mu_i$ and variance $\sigma_i$. Since finding exact solutions for

$$p(\pi|S) = \frac{p(\pi)p(S|\pi)}{\int p(\pi)p(S|\pi)d\pi}$$

is prohibitively expensive to compute, we approximated only the most likely values of $\pi$, denoted $\phi$, using a free-energy approach. This involved optimising the free-energy

$$F = \ln f(\phi; \mu_i, \sigma_i) + \ln f(S; \phi, \sigma)$$

where $f$ denotes the density. We modeled this as a simple set of nodes comprising the current estimate $\phi$, an error term for sensory inputs $\varepsilon_S$, and an error term for priors $\varepsilon_p$, evolving as

$$\dot{\phi} = \varepsilon_S - \varepsilon_p$$
$$\dot{\varepsilon}_S = S - \phi - \Sigma_S \varepsilon_S$$
$$\dot{\varepsilon}_p = \phi - \mu_i I - \Sigma_p \varepsilon_p$$

with parameter updates obtained via gradient ascent:

$$\frac{\partial F}{\partial \mu_i} = I\varepsilon_p, \quad \frac{\partial F}{\partial \Sigma_S} = \frac{1}{2}(\varepsilon_S^2 - \Sigma_S^{-1}), \quad \frac{\partial F}{\partial \Sigma_p} = \frac{1}{2}(\varepsilon_p^2 - \Sigma_p^{-1})$$

For further details on derivations, see [24].

Before the experiment, $\mu_i$ was initialised to the mean of the full GloVe space, and variances $\Sigma_p$ and $\Sigma_S$ were drawn from a uniform distribution. After each trial, posterior estimates were updated via Euler's method based on the semantic embedding of the word the participant reported. This approach was applied separately to both speaker-invariant and speaker-specific priors, with the former modelled using a single speaker across all contexts. Crucially, whenever prior estimates were used outside this estimation procedure, they were based on the posterior of the previous trial within that context to prevent incorporation of future information.

### 4.6 Modeling choice behaviour

Behavioural responses from experiments one and two were analysed in R (version 4.03) [91] using lme4 [92] and emmeans [93]. Generalised linear mixed models were fit using the 'bobyqa' optimiser with a maximum of $2e^5$ iterations. To increase generalisability, a maximal model fitting procedure was followed [94,95] to identify the maximal random effects structure that allowed the model to converge without singular fit. Residuals within the identified maximal model were inspected using DHARMa [96].

Behavioral choices were modeled as

$$p(\text{chose word}) \sim t \times \kappa + t \times p(\text{word}|\text{speaker}) + (1|\text{participant})$$

where trial number $t$, perceptual bias $\kappa$ and the probability of the word given the speaker $p(\text{word}|\text{speaker})$ were scaled to have zero mean and unit variance. Note that probabilities of the word given the speaker were computed as the cosine similarity between the word's embedding and the priors (see Estimating real-time semantic models). For the random effects structure, we tested all reasonable permutations that included these predictors as well as the position of the word on screen, the semantic context, the speaker, and their distinguishing visual feature.

The maximal identifiable model included the following random effects structure for data from experiment one:

$$(0 + \kappa + p(\text{word}|\text{speaker})|\text{participant})+$$
$$(1|\text{speaker} \times \text{context}_1 \times \text{context}_2)+$$
$$(1|\text{participant} \times \text{context}_1 \times \text{context}_2)$$

and for data from experiment two:

$$(1 + t + \kappa + p(\text{word}|\text{speaker})|\text{participant} \times \text{context}) +$$
$$(1|\text{face} \times \text{feature})$$

Note that, while random effects structures differ slightly between experiments, all maximal models included the critical random slopes by $\kappa$ and $p(\text{word}|\text{speaker})$.

## 4.7 Preprocessing EEG data

EEG data were preprocessed using MNE [97] following a preregistered [26], standard preprocessing pipeline [98]. Line noise was removed with a notch filter at 50$Hz$ and harmonics. Channels were demeaned and visually inspected for excessive noise. Noisy channels were interpolated ($N = 0 \pm 1$). To identify rare excessive muscle activity, data were bandpass filtered (110-140$Hz$, IIR filter), epoched from -500 to 2500$ms$, z-scored, and visually inspected. Bad trials were flagged. EEG data were then bandpass filtered (0.5-50$Hz$, IIR filter) and epoched (-500 to 2500$ms$ relative to audio onset). Independent component analysis [99] was employed to remove ocular, cardiac, and muscle-related artifacts. Epochs were re-referenced to the global average and underwent a final visual inspection for trial selection (trials dropped in part one: $N = 8 \pm 6$, part two: $N = 5 \pm 3$). Crucially, this trial selection was applied only for single-trial EEG encoding models where severe artifacts would degrade measured encoding quality. To preserve the critical structure of identical morphs being perceived given different speaker contexts (see Fig 1), no trials were dropped in the cRSA regression approach. For single-trial modeling of EEG responses, we applied an additional lowpass filter ($\leq 15Hz$, IIR filter) [15]. All data were downsampled to 200$Hz$ to speed-up subsequent analysis steps.

## 4.8 Stimulus reconstruction of morphs

To probe the content of low-level sensory representations, we decoded gammatone spectrograms from neural responses following presentation of a morph. To do this, gammatone spectrograms were extracted from raw audio signals at 200$Hz$ across 28 logarithmically spaced frequency bands between 50–11025$Hz$, analogous to previous work [100]. To improve estimation, spectrograms were smoothed with a 100$ms$ boxcar kernel [101].

We then trained subject-level multivariate stimulus reconstruction models [19] mapping from neural data $X$ to gammatones $s$:

$$\hat{s}(t) = \sum_{n} \sum_{\tau} r(t + \tau, n)g(\tau, n)$$

Here, $\hat{s}(t)$ is the reconstructed gammatone representation at time point $t$, $r(t + \tau, n)$ is the neural response $X$ at $t$ and channel $n$ time-lagged by $\tau$, and $g(\tau, n)$ is the spatial filter at lag $\tau$. We used a time window of $\tau = \{0, ..., 250\}ms$. We solved for spatiotemporal filters $G$ using sklearn [102], sklearn-intelex [103] and PyTorch [77]:

$$\arg \min_{G} \sum_{t} (s_t - G^T R_t)^2 + \alpha_G ||G||^2$$

where optimal ridge penalties $\alpha_G$ were found using leave-one-out cross-validation to test 20 logarithmically spaced values between $1e^{-5}–1e^{10}$. Inputs $X$ and $s$ were standardised to have zero mean and unit variance. Models were estimated using 5-fold cross-validation. Out-of-sample performance was measured as the Pearson correlation between real spectrograms $s$ and reconstructed spectrograms $\hat{s}$ in the held-out test set, averaged over frequency bins and time points. Decoded patterns were computed following [104]. To improve robustness, this procedure was repeated over 100 permutations of the data. Final estimates were obtained by averaging out-of-sample performance, decoded patterns and stimulus reconstructions.

**Statistical inference.** To determine overall reconstruction performance, we performed a two-sided one-sample t-test over performance averaged over frequency bands. Further, we tested reconstruction performance in individual frequency bands using two-sided one-sample t-tests. Bonferroni-Holm corrections were applied to adjust for multiple comparisons.

## 4.9 Modeling acoustic expectations as top-*k* predictions

Our experimental design was tailored to address the question: How do expectations shape neural representations? Principally, a morph *sea-tea* given speaker context *nature* may be represented more like *sea* than *tea*, or vice versa. However,

it is unlikely that the brain makes predictions that are this concrete, as it would require, for example, predicting *sea* at trial *t* (*sea-tea|nature*) but immediately thereafter *kayak* at trial *t* + 1 (*kayak-cognac|nature*). In reality, the brain is assumed to make distributional predictions instead [1]. For example, the distributional expectation for *nature q*(*nature*) may be:

$$q(nature) = \begin{bmatrix} p(sea|nature) \\ p(kayak|nature) \\ \vdots \\ p(tea|nature) \end{bmatrix} = \begin{bmatrix} 0.14 \\ 0.10 \\ \vdots \\ 0.02 \end{bmatrix} \tag{1}$$

where $p(word_i|context)$ is computed from semantic embeddings of individual words $e_i$ and semantic priors obtained from free-energy models $\pi$:

$$p(w_i|\pi) = \frac{1}{2\bar{p}}\left(\frac{e_i \cdot \pi}{||e_i||\,||\pi||} + 1\right) \qquad \text{where} \qquad \bar{p} = \sum_{i=0}^{k} p(w_i|\pi)$$

In other words, the probability of any word given a context is the normalised cosine similarity $\mathcal{Z} \in [0, 1]$ between the semantic embedding of that word and the semantic embedding of the context obtained from free-energy models that evolve over the experiment, measuring their probability as a function of how similar a word is to a given prior. Note that, following prior work [37], we use cosine similarity rather than the likelihood of $e_i$ under $\mathcal{N}(\mu_\pi, \sigma_\pi^2)$ as a computational simplification that works particularly well in high-dimensional semantic embeddings. Note also that context subsumes not only speaker-specific priors, but also speaker-invariant priors (i.e., context across the experiment).

Generally, we may then obtain an estimate of the prediction as:

$$\mathbb{E}_f(\pi) = \sum_{i=0}^{N} p(w_i|\pi)f(w_i) \qquad \text{where} \qquad \sum_{i=0}^{N} p(w_i|\pi) = 1$$

where *f* describes some feature we are interested in (for example, acoustic spectrograms), $\pi$ refers to the speaker-specific or speaker-invariant semantic prior we are applying, and $w_i$ is word *i* of *N* total words.

While this simplifies the prediction problem by assuming only a fixed vocabulary over stimuli included in the study, we can be reasonably certain that these truly reflect expected words. This is because, during stimulus creation, categories in semantic space were sampled densely, including those words that most closely aligned with any category. Therefore, these words should inherently reflect the most predictable words given any context.

When modeling semantic feature predictions, this approach could be approximated by comparing a morph *sea-tea* to its constituent words because semantic contexts reflect strong categorical structure whereby any word in *nature* is more similar to other words of *nature* than other contexts. Critically, words that are similar in meaning can be highly heterogeneous in acoustic features and vice versa (e.g., *dog-canine* or *bear-bare*). Consequently, modeling acoustic expectations requires a more explicit commitment to distributional predictions.

Therefore, here we model acoustic predictions as:

$$\mathbb{E}_s(\pi) = \sum_{i=0}^{k} p(w_k|\pi)s(w_k)$$

where *s* refers to spectrograms of individual words. While the resulting acoustic template $\mathbb{E}_s(\pi)$ may not correspond to a naturalistic waveform, the use of non-negative spectrogram magnitudes ensures that there is no destructive interference: Energy patterns of different words are mixed rather than cancelled, highlighting the spectral regularities between words as a function of their probability.

To illustrate the conservation of energy here, consider a decoder with a linear decision function:

$$f_c(x) = w_c^T x + b_c$$

with the expected top-*k* template:

$$\bar{x}_\pi = \mathbb{E}[x|\pi] = \sum_w p(w|\pi) x_w$$

Then for every feature *c*:

$$
\begin{aligned}
f_c(\bar{x}_\pi) &= w_c^T \left( \sum_w p(w|\pi) x_w \right) + b_c \\
&= \sum_w p(w|\pi) w_c^T x_w + b_c \\
&= \sum_w p(w|\pi)(w_c^T x_w + b_c) + \left( 1 - \sum_w p(w|\pi) \right) b_c \\
&= \sum_w p(w|\pi) f_c(x_w) \qquad \text{since} \qquad \sum_w p(w|\pi) = 1
\end{aligned}
$$

In other words, decoding features from top-*k* predictions (probability-weighted spectrogram templates) is formally equivalent to decoding features from each of the *k* individual words, weighted by their probabilities. This equality holds even with intercepts and affine transformations. An empirical validation thereof as well as an extension to representational similarity is provided in a Jupyter notebook (topk_spectrogram_validation) available on GitHub [105].

Consequently, this approach allows us to create acoustic templates that reflect statistics of expected acoustic features that are predictable given a semantic prior. One caveat with this approach is the implicit simplifying assumption that words are predicted in full. Contrary to this assumption, we would expect the brain to make highly transient and auto-regressive predictions that are updated as the acoustic signal unfolds. For example, if we originally predict <crave>, <crane>, and <crowd>, once /kɹeɪ/ has been heard, <crowd> should be neglected–and predictions should be reweighted accordingly. This simplification was made for computational tractability.

### 4.10 Regressing composite representational similarity

To determine whether semantic expectations influenced the representation of morphed words at the acoustic level, we computed within-item composite representational similarity matrices [20] comparing reconstructed spectrograms based on the recorded EEG data of the morphed stimuli with the real spectrograms of the audio files making up the respective morphs (e.g., /siː/ and /tiː/ for morph /‑iː/). The observed cRSM for each morph *n* at time point *t* was defined as:

$$\mathrm{cRSM}(n, t) = \begin{bmatrix} f(/‑iː/|\mathrm{nature}, /\mathrm{siː}/) & f(/‑iː/|\mathrm{food}, /\mathrm{siː}/) \\ f(/‑iː/|\mathrm{nature}, /\mathrm{tiː}/) & f(/‑iː/|\mathrm{food}, /\mathrm{tiː}/) \end{bmatrix}$$

where *f* denotes cosine similarity computed over the frequency bins of gammatone spectrograms. This yielded a cRSM of size $120 \times 2 \times 2 \times 200$ (morph pairs × contexts × words × time points) per participant.

Note that this departs from conventional RSA [20] where RSMs are computed between items, which would have yielded an RSM of size $240 \times 240 \times 200$ (morphs × morphs × time points). Here, we opted for this approach because our research question and design pose a contrast at the within-item level: What is the direction in which predictions change

the sensory representations of the same acoustic morph given different speaker contexts–towards or away from predictions (see Fig 2)? Representational similarity allowed us to jointly model the influence of predictions in different modalities (i.e., acoustics and semantics), while anchoring decoded neural representations directly to real words allowed us to test this directly, without strong mathematical commitments to specific formulations underlying these computations that remain debated [8] but would have been required in a conventional second-order comparison through RSA [20]. Converging evidence from conventional RSA is presented in the supplement (see Conventional representational similarity analysis, S7 Fig, S8 Fig).

We then generated hypothesis cRSMs to assess which representational structures best explained the observed cRSMs. We formulated three core hypotheses to test how acoustic representations changed as a function of the speaker context (see Fig 2B). For each prior-dependent hypothesis, predictions were derived from both speaker-specific and speaker-invariant priors. Both speaker-specific and -invariant hypotheses were therefore computed following the same procedure, with the only difference being the exact semantic prior distribution $\pi$, derived from speaker-specific or speaker-invariant free-energy models (see Estimating real-time semantic models), that was used.

**Prior-independent morph-based representation.** First, as a baseline, if presented morphed words are processed based on the pure, i.e., expectation-free, acoustics of the morph, representational similarity should reflect the direct comparison between the morph's gammatone spectrogram (e.g., $/\text{\underline{i:}}/$) and the respective real spectrogram making up the morph (e.g., $/\text{si:}/$ and $/\text{ti:}/$).

**Prior-dependent acoustic representation.** If semantic priors influence the processing of the morphed words already at an acoustic stage, within-item similarity should reflect expected acoustics derived from the semantic priors. We estimated these expectations by computing the top-$k$ predicted words:

$$\arg\max_{\text{word}} \sum_{i}^{k} p(\text{word}_i|\pi)$$

where $\pi$ is a speaker-specific or -invariant semantic prior distribution derived from free-energy models (see Estimating real-time semantic models). Normalised probabilities were then used to compute a prior-weighted template from the top-$k$ words that reflected expected acoustic features:

$$s(\text{predicted}|\pi) = \sum_{i}^{k} p(\text{word}_i|\pi)s(\text{word}_i)$$

Since spectrogram magnitudes are inherently non-negative, there is no destructive interference here. Instead, this weighted procedure aggregates spectral energy from individual word predictions that together form an expected template that reflects predicted acoustic features (see Modeling acoustic expectations as top-k predictions, Regressing composite representational similarity; see also S6 Fig). Expected acoustic templates were then compared with the respective real spectrograms making up the morph (e.g., $/\text{si:}/$ and $/\text{ti:}/$).

Because language exhibits strong baseline predictability [3,37], we controlled for general acoustic expectations that participants may have about German. To do this, we repeated the exact procedure for top-$k$ predictions outlined above, but generated predictions from non-specific priors sampled from an isotropic Gaussian around the vector space's centroid at every trial. Across large numbers of trials and $k$, this therefore approximates general statistical regularities of German, allowing us to control for general predictive processes that were irrespective of our experiment.

**Prior-dependent semantic representation.** If semantic priors influence the processing of the morphed words at higher-level processing stages—for example, shifting sensory representations uniformly per semantic context—composite representational similarity should reflect direct comparisons between the prior distribution of the semantics given the

speaker $\pi$(speaker) or speaker-invariant prior $\pi$ (see Estimating real-time semantic models) and the word embeddings of the words making up the morph (e.g., *sea* and *tea*). As before, we computed an additional control predictor to capture more global statistical regularities.

**Modeling approach.** To test whether speaker context shifted participants' perception of the same acoustic morphs, we used ridge regressions mapping from hypothesis cRSMs $X$ to sensory cRSMs $y$ for each participant:

$$\hat{y}(t) = \beta_t X_t + \epsilon$$

where $\beta_t$ are the coefficients at time point $t$. We constructed a temporally expanded design matrix $D$ and temporally expanded outcomes $Y$, and solved for the full model $\beta$ using sklearn [102], sklearn-intelex [103] and PyTorch [77]:

$$\arg\min_{\beta} \sum_i (Y_i - \beta^T D_i)^2 + \alpha_\beta ||\beta||^2$$

Ridge penalties $\alpha_\beta$ were optimised using leave-one-out cross-validation to test 20 logarithmically spaced values between $1e^{-5}$–$1e^{10}$ [102]. Models were estimated using 5-fold cross-validation with 50 repetitions [106] and all outcomes and predictors were scaled to have zero mean and unit variance based on the training set to prevent data leakage. Out-of-sample performance was defined as the Pearson correlation between observed sensory cRSMs $y(t)$ and predicted sensory cRSMs $\hat{y}(t)$ in the held-out test set [100,101]. To ensure comparability across models, we normalised coefficients as:

$$\hat{\beta}_{j,t} = \frac{\beta_{j,t}}{\max_t \sum_j |\beta_j|}$$

Thus, $\hat{\beta}_j$ represents the relative contribution of predictor $j$ to the variance explained [107], while meaningfully preserving signs.

**Candidate models.** Baseline models predicted sensory cRSMs from hypothesis cRSMs based on audio spectrograms of morphs as well as audio spectrograms and semantic embeddings from general language-specific expectations. We then performed step-wise inclusion of speaker-specific and speaker-invariant acoustic and semantic hypothesis cRSMs.

**Statistical inference.** Improvements in out-of-sample predictions for each hypothesis cRSM were tested using two-sided paired t-tests, with Bonferroni-Holm corrections applied for multiple comparisons. Significance of temporally resolved variance explained by $\hat{\beta}$ was assessed through cluster-based permutation tests [51]. To compensate for the number of predictors, Bonferroni corrections were applied.

### 4.11 Validating top-*k* acoustic predictions

If speaker-specific acoustic top-*k* predictions truly captured meaningful acoustic expectations of participants, disrupting these predictions should similarly deteriorate model performance.

To test this, we refit speaker-invariant and speaker-specific semantic priors (see Estimating real-time semantic models) for each participant while randomising their responses. This meant that semantic priors were now fit over the semantically coherent words in only 50% of trials. The remaining 50% of trials fell into one of five distinct alternative contexts (each with 10% chance). Because these were five disparate contexts that, jointly, were not semantically coherent, the newly estimated semantic priors still converged to coherent speaker-specific semantic priors that reflected a plausible estimate of the target context. Critically, however, they no longer represented the true semantic prior of any participant at any point in time.

Because this procedure preserved the overall semantic structure of priors but disrupted the degree to which participant-specific expectations were captured, we expected that model performance should be degraded for speaker-specific

acoustic predictions, but that speaker-invariant acoustic predictions should remain largely unaffected given that experiment-specific statistics remain comparable.

To test this, we generated speaker-invariant and -specific acoustic and semantic top-$k$ predictions from these foil priors. In accordance with our main analysis, we set $k = 5$. We then regressed foil cRSMs on sensory cRSMs, mirroring the procedure from our main findings (see Regressing composite representational similarity). We performed paired samples t-tests between performance of models using true and foil semantic priors with Bonferroni-Holm corrections applied.

### 4.12 Conventional representational similarity analysis

To ensure that our finding of dominant sharpening at the acoustic level did not depend on the choice of analysis method (within-item composite RSM regression) or spectrogram averaging, we also conducted a conventional between-item representational similarity analysis (RSA) [20]. In this analysis we computed full representational similarity matrices (RSMs) for decoded sensory representations and all hypotheses from the original analysis (see Regressing composite representational similarity). Between-item RSA is typically used to compare representational geometries across systems [20]. As previously discussed (see Results), this requires a commitment to specific mathematical formulations of the computations of interest, which is challenging for predictive processing where a particularly high number of potential formulations exist [7,8,31–33]. To reduce degrees of freedom [34–36], we adopted two common mathematical formulations of sharpening and prediction error that are mathematically simplified by assuming no precision weighting [8,14]:

$$\text{sharpening(observation | expectation)} = \text{observation} \times \text{expectation}$$
$$\text{prediction error(observation | expectation)} = \text{observation} - \text{expectation}$$

Sensory RSMs were computed from cosine similarity between reconstructed spectrograms, and hypothesis RSMs were computed from: 1) morph spectrograms, 2) speaker-specific and speaker-invariant top-$k$ acoustic predictions, and 3) speaker-specific and speaker-invariant semantic predictions. For this analysis we set $k = 1$ to avoid averaging across spectrograms. For semantic RSM computations, sharpening and prediction error formulations required disambiguation of morphs towards the target word (e.g., observation(sea-tea|nature) = sea) that were paired with their respective semantic priors (e.g., expectation(sea-tea|nature) = $\pi$(nature)); see Estimating real-time semantic models). In all, this yielded RSMs of size $240 \times 240 \times 200$ (morphs × morphs × time points) of which the upper triangles were used for regressions ($28,680 \times 200$).

To assess the influence of individual hypothesis RSMs, we used ridge regressions mapping from hypothesised RSMs $X$ to observed sensory RSMs $y$ for each participant:

$$\hat{y}(t) = \beta_t X_t + \epsilon$$

where $\beta_t$ are coefficients at time point $t$, $X_t$ are hypothesis RSMs and $\hat{y}$ are predicted sensory RSMs. We solved for models $\beta$ using PyTorch [77] and MVPy [108]:

$$\arg\min_{\beta_t} \sum_i (y_i - \beta_t^T X_t)^2 + \alpha_\beta ||\beta_t||^2$$

Ridge penalties $\alpha_\beta$ were optimised using leave-one-out cross-validation, testing 20 logarithmically spaced values between $1e^{-5} - 1e^{10}$ [102]. All models were estimated with 5-fold cross-validation and 50 repetitions. Predictors were scaled to have zero mean and unit variance, based on training sets to prevent data leakage. Model performance was measured as out-of-sample Pearson correlation between $y(t)$ and $\hat{y}(t)$ in the held-out test sets, mirroring the procedure followed in our main analysis (see Regressing composite representational similarity).

Out-of-sample prediction performance between models was compared using two-sided paired t-tests with Bonferroni-Holm corrections, mirroring the original model comparison procedure in cRSM regressions (see Regressing composite representational similarity). For results from this analysis, see S7 Fig and S7 Table. For results from this analysis while restricting the temporal window to that of the shortest possible prediction (0.0–0.48$s$), see S8 Fig and S8 Table.

### 4.13 Pretrained transformers as statistical surrogates

For modeling of single-trial EEG responses, we required estimates of speaker-prior independent surprisal that are known to modulate neural responses [15]. Since phonotactic, lexical, and semantic surprisal are undefined for morphed words, we assessed how well these properties were captured by activations of pretrained transformers in a separate analysis.

To achieve this, an excerpt of a larger narrative (Alice in Wonderland, produced by the same speaker)–wherein phonotactic, lexical, and semantic surprisal were well-defined–was fed into 'Wav2Vec2.0-large-xlsr-53-german' [23,109] using PyTorch [77] and Transformers [110]. Layer-specific activations were extracted for narrative data. For computational efficiency, we performed layer-wise principal component analysis over narrative data and projected activations onto a 5-dimensional subspace. For layer selection and to confirm that the resulting activations included the relevant information theoretic measures, we built back-to-back decoding models [41]. In a first step, we built models mapping from layer activations $X$ to observed measures $y$ through a filter $G$:

$$\arg\min_{G} \sum_i (y_i - G^T X_i)^2 + \alpha_G ||G||^2$$

where $y_i$ is a feature at trial $i$, $X_i$ is the corresponding layer-specific activation and $\alpha_G$ is a regularisation parameter. These models were fit over the first half of the narrative data. The remaining half was then used to find the unique variance explained by each feature by mapping from decoded features $\hat{y}$ to observed features $y$ through a filter $H$:

$$\arg\min_{H} \sum_i (y_i - H^T \hat{y}_i)^2 + \alpha_H ||H||^2$$

In both cases, we solved for filters $G$ and $H$ using sklearn [102] and PyTorch [77]. Optimal ridge penalties $\alpha_G$, $\alpha_H$ were found using leave-one-out cross-validation to test 20 logarithmically spaced values between $1e^{-5}$–$1e^{10}$ [102]. This approach was employed for all time points from $0ms$-$500ms$ following phoneme onset, averaging over time to obtain an estimate of the causal contribution of each feature. To improve robustness, this procedure was repeated for 50 permutations of the narrative data [106]. Finally, we computed the proportion of variance explained by each feature to facilitate interpretation [107]:

$$R^2 = \frac{H}{\max_L \sum_i^F H_i}$$

where $L$ are all layers of the transformer and $F$ refers to all features. To identify the layer that was maximally sensitive to joint surprisal measures, we then ranked all layers per feature and took the geometric mean across features. The best layer was then selected by taking the median across permutations. For full results, please see S9 Fig. Features were operationalised as follows:

**Phonotactic surprisal.** The probability of a phoneme given its preceding context was estimated using CLEARPOND [111]. At each word position, all phonotactically consistent words (e.g., /t/: *tea*, *tear*, *trousers*, ...) were identified, and the probability of the next phoneme (e.g., $/\text{iː}/$) given all observed next phonemes $x$ was computed as:

$$p(\text{phoneme}) = N^{-1} \sum_i^N \mathbb{1}(x_i = \text{phoneme})$$

**Lexical surprisal.** The probability of a word was estimated using its relative frequency *f* in CLEARPOND [111] as $p(\text{word}) = \log_{10} f$.

**Semantic surprisal.** The probability of a word was derived from the cosine similarity of its GloVe embedding *g* relative to the mean embedding $\mathbb{E}[G]$:

$$p(\text{word}) = 1 + \frac{g \cdot \mathbb{E}[G]}{||g|| \; ||\mathbb{E}[G]||}$$

For all features, surprisal was defined as $-\log_2 p$. Results revealed that, across an array of *d*-dimensional projections of model activations, layer 19 (transformer layer 12) was consistently sensitive to prior-independent phonotactic, lexical and semantic surprisal (see S9 Fig).

Finally, we fed all morphs into the model and extracted model activations at layer 19. Model activations were projected into 5- and 10-dimensional subspaces and stored for later use as surrogate predictors that jointly controlled for measures of surprisal that were otherwise undefined for morphs when modeling single-trial EEG responses.

### 4.14 Encoding of single-trial EEG data

To investigate whether participants showed neural signatures of speaker-specific acoustic or semantic surprisal, we built encoding models [19,21,22] mapping from stimulus features *X* to the recorded neural data *y* through multivariate temporal response functions (mTRFs) for each participant:

$$r(t, n) = \sum_{\tau} w(\tau, n)s(t - \tau) + \epsilon(t, n)$$

Here, $r(t, n)$ is the reconstructed neural signal in channel *n* at time point *t*, $w(\tau, n)$ are the $\beta$-estimates at lag $\tau$, $s(t - \tau)$ are the corresponding stimulus features, and $\epsilon(t, n)$ are the residuals. For $\tau$, we chose a window between $-100ms$ and $800ms$. Continuous predictors (e.g., acoustic envelopes, wav2vec2.0 activations) were averaged within each linguistic unit to create discrete predictors at the phoneme and word levels. For features without explicit segmentation (e.g., wav2vec2.0 activations), we constructed both phoneme- and word-level spike regressors in this way. We solved for the forward model *W* using sklearn [102] and PyTorch [77]:

$$\arg \min_{W} \sum_{t} (y_t - W^T S_t)^2 + \alpha_W ||W||^2$$

where optimal ridge penalties $\alpha_W$ were found using leave-one-out cross-validation to test 20 logarithmically spaced values between $1e^{-5}$–$1e^{10}$. All models were estimated using 5-fold cross-validation with 50 repetitions [106] and all outcomes and predictors were scaled to have zero mean and unit variance based on the training set to prevent data leakage. For predictors, standardisation was applied after lag expansion (i.e., each column of the lagged design matrix $S_t$ was *z*-scored within the training fold) to ensure that ridge regularisation penalised all lags comparably. Out-of-sample performance was defined as the Pearson correlation between observed neural signals $y(t, n)$ and reconstructions thereof $r(t, n)$ in the held-out test set [19].

**Candidate models.** Baseline models included predictors for acoustic envelopes

$$\text{envelope}(t) = \sum_{i}^{F} |s(i, t)|^{0.6}$$

where $F$ are all frequency bands and $t$ refers to time points in gammatone spectrogram $s$, as well as acoustic edges [112]

$$\text{edge}(t) = \sqrt{\frac{1}{2\Delta_t} \int_{t-\Delta_t}^{t+\Delta_t} \left(\text{envelope}(\tau) - \mathbb{E}[\text{envelope}(\tau)]\right)^2 d\tau}$$

where $\mathbb{E}[\text{envelope}(\tau)]$ is the mean envelope over $t \pm \Delta_t$. Baseline models also included predictors controlling for general phonotactic, lexical and semantic surprisal. Generally, control predictors were derived from pretrained transformer activations (see Pretrained transformers as statistical surrogates). However, to rule out the possibility of performance of pretrained transformers given morphs confounding results, we also conducted a second set of analyses where control predictors were computed as direct measures of phonotactic, lexical and semantic surprisal by disambiguating the morph to the target word.

We then performed step-wise inclusion of speaker-specific and speaker-invariant acoustic and semantic surprisal measures, derived from the predicted acoustic $s(\text{predicted}|\text{speaker})$ and semantic features $p(\text{speaker})$:

$$\text{surprisal}_{\text{acoustic}} = -\log_2\left[1 + f(s(\text{stimulus}), s(\text{predicted}|\text{speaker}))\right]$$
$$\text{surprisal}_{\text{semantic}} = -\log_2\left[1 + f(g, p(\text{speaker}))\right]$$

where $f$ denotes cosine similarity and $g$ represents the target word embedding.

Predictors were entered as impulse responses at phoneme or word onsets, depending on feature type. Projected transformer activations, lacking explicit linguistic segmentation, were entered as both.

**Statistical inference.** Encoding performance improvements were tested using two-sided paired t-tests, with Bonferroni-Holm corrections applied for multiple comparisons. Specifically, we tested for within-participant differences between models, e.g. t(baseline, speaker-specific acoustic). Encoding performance of baseline models–measured as the Pearson correlation between observed neural responses $y$ and reconstructions thereof $r$–was confirmed to be above chance using one-sample t-tests (main task: $M = 0.18$, $s.e. = 0.01$, $t(34) = 26.48$, $p = 2.80e^{-24}$; exemplar task: $M = 0.15$, $s.e. = 0.01$, $t(34) = 20.92$, $p = 5.36e^{-21}$).

### 4.15 Estimating spatiotemporal contributions of predictors

To obtain a clearer view of the spatiotemporal contributions of the predictors of interest, we employed a knock-out procedure.

We first estimated the performance of the full model, $r_{\text{full}}$, by computing the Pearson correlation between the observed EEG signals $y(t, n)$ and reconstructed signals $\hat{y}(t, n|S)$, where $S$ is the full time-resolved design matrix (see Encoding of single-trial EEG data). To isolate the influence of a given predictor at a specific time lag $\tau$, we generated a knock-out design matrix $S_\tau$ in which the corresponding coefficient at $\tau$ was set to zero. We then computed the knock-out model's performance, $r_{\text{knock-out}}$, as the correlation between observed signals $y(t, n)$ and reconstructed signals $\hat{y}(t, n|S_\tau)$. The spatiotemporal contribution of the predictor at each lag was quantified as:

$$\Delta r_\tau = r_{\text{full}} - r_{\text{knock-out}}$$

To ensure robustness, models were trained using 5-fold cross-validation and evaluated over 50 permutations of the data. The estimated spatiotemporal contributions $\Delta r_\tau$ were then averaged across folds and permutations. Finally, we normalised these contributions relative to the improvement of the full model over the baseline model's performance, $r_{\text{baseline}}$,

yielding a spatiotemporally resolved measure of the variance explained:

$$R^2_\tau = \frac{\Delta r_\tau}{\mathbb{E}[r_{\text{full}} - r_{\text{baseline}}]}$$

**Statistical inference.** Significance of spatiotemporal variance explained was tested using cluster-based permutation tests [51]. To compensate for the number of coefficients within each model, Bonferroni corrections were applied.

### 4.16 Modeling behaviour in congruent and incongruent trials

Reaction times were analysed in R (version 4.03) [91] using lme4 [92] and emmeans [93]. Linear mixed models were fit with default parameters. A maximal modelling procedure was followed [94]. Residuals of the identified maximal model were inspected visually.

Reaction times were modeled as

$$\log \text{RT} \sim t \times \text{congruence} \times p(\text{word}|\text{speaker}) + (1|\text{participant})$$

where trial number $t$ and the probability of the word given the speaker $p(\text{word}|\text{speaker})$ were scaled to have zero mean and unit variance. For the random effects structure, we included all reasonable permutations of these predictors as well as the position of the word on screen, the semantic context, the speaker and their distinguishing visual feature. The maximal identifiable model included:

$$(1 + t + p(\text{word}|\text{speaker})||\text{participant})$$
$$+ (1 + p(\text{word}|\text{speaker})||\text{context})$$
$$+ (1 + p(\text{word}|\text{speaker})||\text{stimulus})$$
$$+ (1 + p(\text{word}|\text{speaker})||\text{position})$$

## Acknowledgments

We thank Kristin Derben for assistance with recruitment and data acquisition; Jan Ostrowski and Marike Christiane Maack for help with data acquisition; Annika Garlichs, Carina Ufer, Janika Becker, Maï-Carmen Requena-Komuro, and Philipp Schumann for valuable discussions and code review; and Antoniya Boyanova, Arjen Alink, and Ivana Tanasic for their feedback on the study and manuscript.

## Ethics approval

The research was approved by the local ethics committee (Ethics Committee of the Chamber of Physicians in Hamburg, approval no. PV7210).

## Supporting information

**S1 Fig. Results from behavioural analysis.** Summary of key behavioural results from online experiment one (top row) and EEG experiment two (bottom row). In both experiments, participants initially relied on remaining acoustic properties (i.e., morph bias to one of the words within each word pair), but decreased this reliance over time (left). Participants increasingly relied on the probability of the word given the speaker instead (right). Lines represent means, with shaded

areas representing 95%-confidence intervals. For details, see S1 Table and S2 Table. Data and code supporting these findings are available from https://doi.org/10.17605/OSF.IO/SNXQM.
(TIFF)

**S2 Fig. Stimulus reconstruction by frequency band.** Accuracy of stimulus reconstruction models [19] in all 28 individual frequency bands. Bold dots indicate means, with 95%-confidence intervals around them. Small dots represent individual participants. Inlaid topographies show the decoded pattern for this frequency band. ***, **, * represent $p \leq 1e^{-3}$, $p \leq 1e^{-2}$ and $p \leq 5e^{-2}$, respectively. For details, see S3 Table. Data and code supporting these findings are available from https://doi.org/10.17605/OSF.IO/SNXQM.
(TIFF)

**S3 Fig. Influence of the number of predicted words on cRSA regression performance.** We systematically refit models with increasing $k$ to test whether the brain makes a specific number of predictions $k$. Big dots represent group means, with 95%-confidence intervals around them. Small dots indicate individual participants. Big black bars on top indicate $p \leq 5e^{-2}$ for any contrast. For more details, see S4 Table. Data and code supporting these findings are available from https://doi.org/10.17605/OSF.IO/SNXQM.
(TIFF)

**S4 Fig. cRSM regressions using top-19 predictions.** Similarity encoding models using $k = 19$ predictions, because increasing the number of predictions $k$ yielded significant improvements in performance. Left: Results show that both speaker-invariant and speaker-specific acoustic predictions improve model performance, with the best model incorporating both at once. Critically, purely semantic predictions failed to improve performance. This is in line with our results based on $k = 5$, reported in Fig 2. Big dots represent group means, with 95%-confidence intervals around them. Small dots represent individual participants. ***, **, * indicate $p \leq 1e^{-3}$, $p \leq 1e^{-2}$, and $p \leq 5e^{-2}$, respectively. The downward-facing triangle marks the best model overall. Big black bars between groups indicate $p \leq 5e^{-2}$. Right: Coefficients for speaker-specific acoustic predictions showed, again, that there was significant sharpening of neural representations, with a cluster between 0ms-1000ms. Lines indicate mean, with shaded areas around them representing 95%-confidence intervals. Big black lines indicate $p \leq 5e^{-2}$. Data and code supporting these findings are available from https://doi.org/10.17605/OSF.IO/SNXQM.
(TIFF)

**S5 Fig. Correlations between predictors in cRSA regression.** Correlations between all hypothesis cRSMs used in regressing sensory cRSMs for top-$k$ predictions where $k = 5$. Hypothesis cRSMs included baseline morphs (BM), baseline acoustic predictions (BA), baseline semantic predictions (BS), speaker-invariant acoustic predictions (IA), speaker-specific acoustic predictions (SA), speaker-invariant semantic predictions (IS), and speaker-specific semantic predictions (SS). Baseline predictors were included in all models to control for acoustic properties of the morph (BM) as well as general acoustic (BA) and semantic predictions (BS) that were irrespective of our experiment. Left: Correlations were computed between all hypothesis cRSMs and averaged over time points. Individual cells contain means and standard errors computed over subjects. Right: Correlations were computed between all hypothesis cRSMs, but not averaged over time points. In each cell, time points (0.0-1.0s) are plotted on the x-axis, whereas correlations (-1 - 1) are plotted on the y-axis. Per cell, we plot chance level (dashed lines) and mean over subjects (solid lines) with confidence intervals indicated by shaded areas around means. Data and code supporting these findings are available from https://doi.org/10.17605/OSF.IO/SNXQM.
(TIFF)

**S6 Fig. Top-$k$ predictions capture meaningful acoustic expectations.** Validation analysis using foil priors that disrupted participant-specificity of semantic priors. To confirm that performance of top-$k$ acoustic predictions depended on

the trial-specific top-$k$ predictions, we compared performance of predictions from the true speaker-invariant and -specific semantic priors with coherent but not trial-specific foil semantic priors in cRSM regressions (see Validating top-$k$ acoustic predictions). In accordance with our main analysis, we set $k = 5$. As expected, this confirmed that speaker-specific acoustic predictions depended on the trial-specific predictions, indicating that they captured meaningful acoustic predictions. Note that, for visual clarity, we restricted the y-axis to a smaller range. Individual bars represent group means, with black lines indicating 95%-confidence intervals around the mean. Big bars between groups indicate $p \leq 5e^{-2}$. Data and code supporting these findings are available from https://doi.org/10.17605/OSF.IO/SNXQM.
(TIFF)

**S7 Fig. Conventional RSA corroborates findings from cRSA.** Between-item RSA regression models using $k = 1$ predictions to validate results using a conventional approach and without spectrogram averaging (see Conventional representational similarity analysis). Results confirm the dominance of acoustic sharpening (sh) over acoustic prediction errors (pe), as well as the dominance of acoustic over semantic sharpening. Unlike the within-item cRSM regression, this approach also shows modest additive variance explained by both acoustic prediction errors and semantic sharpening and prediction errors, though variance uniquely attributable to either is comparatively smaller (see S7 Table). Therefore, this does not change our interpretation that acoustic sharpening is the dominant computation at the sensory level, but suggests that multiple computations may co-occur at the same level to varying degrees. Individual bars represent group means, with black lines indicating 95%-confidence intervals around the mean. ***, **, * $p \leq 1e^{-3}$, $p \leq 1e^{-2}$, and $p \leq 5e^{-2}$ for model comparisons against baseline, respectively. Big bars between models indicate $p \leq 5e^{-2}$. Data and code supporting these findings are available from https://doi.org/10.17605/OSF.IO/SNXQM.
(TIFF)

**S8 Fig. Length of acoustic predictions does not explain RSA results.** Between-item RSA regression models, setting $k = 1$ to avoid averaging and restricting the temporal window of the analysis to the shortest possible prediction (0.0-0.48$s$) such that differences in length could not affect results (see Conventional representational similarity analysis). Results corroborate the pattern observed without temporal restrictions (see S7 Fig), ruling out prediction length as a confounding factor (see S8 Table). Individual bars represent group means, with black lines indicating 95%-confidence intervals around the mean. ***, **, * indicate $p \leq 1e^{-3}$, $p \leq 1e^{-2}$, and $p \leq 5e^{-2}$, respectively. Big bars between groups indicate $p \leq 5e^{-2}$. Data and code supporting these findings are available from https://doi.org/10.17605/OSF.IO/SNXQM.
(TIFF)

**S9 Fig. Layer selection in pretrained transformer.** To identify the layer within word2vec2.0 that best captured phonotactic, lexical and semantic surprisal, we performed back-to-back decoding [41] over an independent set of narrative stimuli (see Pretrained transformers as statistical surrogates). Here, we show the relative causal contribution of each feature across layers, for a number of PCA projections with dimensionality $d$. Small dots represent means, with 95%-confidence intervals around them. Downward-facing triangles represent the best layer overall for each $d$. Crucially, for all $d$ tested, layer 19 (transformer layer 12) was identified as the most informative layer. This is in accordance with recent results finding that middle layers of transformers tend to reflect neural processing best [3]. Data and code supporting these findings are available from https://doi.org/10.17605/OSF.IO/SNXQM.
(TIFF)

**S10 Fig. Encoding models are robust to increased transformer dimensionality.** To test the robustness of the effect of speaker-specific semantic surprisal, we refit the models using $d = 10$ features from transformer activations. Left: This revealed that, again, only speaker-specific semantic surprisal improved encoding performance—though here results varied more strongly, likely due to the now relatively high number of predictors within the model. Big dots represent group means, with 95%-confidence intervals around them. Small dots represent individual participants. ***, **, * indicate $p \leq 1e^{-3}$, $p \leq 1e^{-2}$, and $p \leq 5e^{-2}$, respectively. The downward-facing triangle marks the best model overall. Big bold lines

indicate $p \le 5e^{-2}$. Right: As before, speaker-specific semantic surprisal explained significant variance, with a cluster across a wide array of sensors around $110ms$-$400ms$. The inlaid topography shows variance explained at each sensor position within the cluster, with channels contributing to the cluster highlighted in black. Lines indicate means, with 95%-confidence intervals as shaded areas around them. Bold black lines indicate $p \le 5e^{-2}$. Data and code supporting these findings are available from https://doi.org/10.17605/OSF.IO/SNXQM.
(TIFF)

**S11 Fig. Results from encoding models are not contingent on transformers.** To further probe the robustness of speaker-specific semantic surprisal, we refit the models using control predictors derived from the word that was more likely under the speaker prior. Consequently, these models were implicitly biased against speaker-specific semantic surprisal, as this disambiguation of the morph meant some speaker-specific information was already encoded in the control models. Left: Again, we find that speaker-specific semantic surprisal was the only predictor that significantly improved model performance. Big dots represent group means, with 95%-confidence intervals around them. Small dots represent individual participants. ***, **, and * represent $p \le 1e^{-3}$, $p \le 1e^{-2}$, and $p \le 5e^{-2}$, respectively. The downward-facing triangle marks the best model overall. Big black lines indicate $p \le 5e^{-2}$. Right: Again, we find that speaker-specific semantic surprisal explains significant variance, with a cluster across all sensors between $-100ms$-$410ms$. The inlaid topography shows variance explained at each sensor position, with sensors contributing to the cluster marked in black. Lines indicate means, with 95%-confidence intervals around them. Big bold lines indicate $p \le 5e^{-2}$. Data and code supporting these findings are available from https://doi.org/10.17605/OSF.IO/SNXQM.
(TIFF)

**S12 Fig. No systematic differences in peak latency of semantic surprisal responses.** Peak latencies of semantic surprisal effects across morph, congruent and incongruent trials, demonstrating that there was no significant difference between peak latencies. Note that these represent raw latency estimates over the temporal extent of the cluster. No jack-knife procedure was applied. Big dots indicate group means, with 95%-confidence intervals around them. Small dots represent individual participants. Data and code supporting these findings are available from https://doi.org/10.17605/OSF.IO/SNXQM.
(TIFF)

**S13 Fig. Overview of stimulus materials. A** Visualisation of the semantic space spanned by the words used in this experiment, as projected into two dimensions using PCA. Dots represent individual words, coloured by the semantic context they were most likely under. **B** Gaussian approximation of the distribution of perceptual bias of morphs across the experiment, showing that stimuli exhibit no systematic bias. **C** Perceptual bias of morphs for individual contexts. Big dots indicate context means, with 95%-confidence intervals around them. Small dots represent individual morphs. Again, no systematic biases exist. Data and code supporting these findings are available from https://doi.org/10.17605/OSF.IO/SNXQM.
(TIFF)

**S1 Table. Behavioural analysis in online experiment.** A generalised linear model revealed that participants reported hearing words as a function of their probability given the speaker prior (*fit*), remaining perceptual differences ($\kappa$), and trial number (*t*).
(PDF)

**S2 Table. Replication of behavioural analysis in EEG experiment.** Effects of the probability of a word given the speaker prior (*fit*), remaining perceptual differences ($\kappa$), and trial number (*t*) on the reported word were replicated in the EEG experiment.
(PDF)

**S3 Table. Stimulus reconstruction per frequency band.** Across all 28 frequency bands of the gammatone filterbank, stimulus reconstruction performance was significantly above chance level.
(PDF)

**S4 Table. Influence of the number of predicted words on cRSA regression performance.** Results from consecutive contrasts of top–$k$ within-item cRSM regression models showed consistent increases in model performance, albeit with diminishing returns for larger $k$.
(PDF)

**S5 Table. cRSM regressions using top-19 predictions.** Results from contrasts in within-item cRSM regression models using $k = 19$ confirmed the pattern reported for $k = 5$.
(PDF)

**S6 Table. Top-$k$ predictions capture meaningful acoustic expectations.** Comparing cRSM regressions using top-$k$ predictions derived from participant-specific and plausible foil priors showed better performance of participant-specific priors (see Validating top-$k$ acoustic predictions). This indicates that they captured meaningful acoustic predictions by participants.
(PDF)

**S7 Table. Conventional RSA corroborates findings from cRSA.** Results from between-item RSA regression models using $k = 1$ (see Conventional representational similarity analysis) mirrored results obtained from cRSA regressions.
(PDF)

**S8 Table. Length of acoustic predictions does not explain RSA results.** Results from between-item RSA regression models using $k = 1$ and restricting the temporal window to $0.0\text{-}0.48s$ to rule out prediction length as a confounding factor (see Conventional representational similarity analysis).
(PDF)

**S9 Table. Encoding models are robust to increased transformer dimensionality.** Results from contrasts in single-trial encoding models using pretrained transformers with $d = 10$.
(PDF)

**S10 Table. Results from encoding models are not contingent on transformers.** Results from contrasts in single-trial encoding models using control predictors derived from target words.
(PDF)

**S11 Table. Double dissociation of semantic congruency and prior specificity.** Results from contrasts in single-trial encoding models in the follow-up task.
(PDF)

**S12 Table. List of word- and context-pairs.** All stimulus pairs along with their corresponding context pairs used in the present study.
(PDF)

## Author contributions

**Conceptualization**: Fabian Schneider, Helen Blank.

**Data curation**: Fabian Schneider.

**Formal analysis**: Fabian Schneider.

**Funding acquisition**: Helen Blank.

**Investigation**: Fabian Schneider.

**Methodology**: Fabian Schneider, Helen Blank.

**Project administration**: Fabian Schneider.

**Resources**: Fabian Schneider.

**Software**: Fabian Schneider.

**Supervision**: Helen Blank.

**Visualization**: Fabian Schneider.

**Writing – original draft**: Fabian Schneider.

**Writing – review & editing**: Fabian Schneider, Helen Blank.

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
