## [Editor Report · Decision Letter 0]

6 Jun 2025

Dear Dr Schneider,

Thank you for submitting your manuscript entitled "Sensory sharpening and semantic prediction errors unify competing models of predictive processing in communication" for consideration as a Research Article by PLOS Biology.

Your manuscript has now been evaluated by the PLOS Biology editorial staff and I am writing to let you know that we would like to send your submission out for external peer review.

Once your full submission is complete, your paper will undergo a series of checks in preparation for peer review. After your manuscript has passed the checks it will be sent out for review. To provide the metadata for your submission, please Login to Editorial Manager (https://www.editorialmanager.com/pbiology) within two working days, i.e. by Jun 10 2025 11:59PM.

Kind regards,

Christian

Christian Schnell, PhD

Senior Editor

PLOS Biology

cschnell@plos.org

---

## [Decision Letter · Decision Letter 1]

1 Aug 2025

Dear Dr Schneider,

Thank you for your patience while your manuscript "Sensory sharpening and semantic prediction errors unify competing models of predictive processing in communication" was peer-reviewed at PLOS Biology. It has now been evaluated by the PLOS Biology editors, an Academic Editor with relevant expertise, and by several independent reviewers.

In light of the reviews, which you will find at the end of this email, we would like to invite you to revise the work to thoroughly address the reviewers' reports.

As you will see below, the reviewers mainly request to improve the clarity of the presentation and provide better justifications for the methodological choices. Reviewer 4 additionally raises a few technical concerns regarding the statistical analyses and the RSA-like approach that will need to be addressed.

Given the extent of revision needed, we cannot make a decision about publication until we have seen the revised manuscript and your response to the reviewers' comments. Your revised manuscript is likely to be sent for further evaluation by all or a subset of the reviewers.

**IMPORTANT - SUBMITTING YOUR REVISION**

*Re-submission Checklist*

*Published Peer Review*

*PLOS Data Policy*

*Blot and Gel Data Policy*

Sincerely,

Christian

Christian Schnell, PhD

Senior Editor

PLOS Biology

cschnell@plos.org

REVIEWS:

Reviewer #1: Ps learn to associate six faces with six contexts (food, fashion, technology, politics, arts, and nature) in an initial phase, where they are shown the face and told a one sentence summary of their interests. In the subsequent phase (EEG), they see a face, which is followed by an ambiguous word (two morphed). In part 2, the morph was replaced by a congruent or incongruent word. They were then shown feedback according to whether they had selected the congruent or incongruent option. Priors were fit to their responses from part 1 concerning the speaker-word associations. They fit the priors either as speaker-specific or speaker-general, i.e., across semantic contexts. Behaviour is biased towards priors. Neurally the picture is more complex. The signal is biased toward speaker-specific acoustic predictions at 165-1000 ms. And biased away from speaker-specific semantic predictions at ~150-630 ms.

The manuscript is well written and the study appears interesting and well-designed. The behavioural effects seem a given in light of their design, but the neural effects are potentially very interesting. The analysis appears clever, although the details are difficult to follow - even as a relative specialist. Clarification is needed throughout. It also felt like different analysis strategies were used to test different hypotheses, and the logic here was hard to follow. Perhaps these were all specified in a prereg (in which case it would need less justification) but currently the reader will presumably need more justification for why each analysis for each question. If these issues could be fixed it would seem to make an interesting contribution to the literature - that neural signals are biased towards more sensory expectations and biased away from more semantic ones.

Specifics:

1. That the design renders the behavioural effects a given. The authors test whether perception is pulled towards the context or pushed away from it. They find it is pulled towards it, and this effect increases over time. But the context-congruent option is highlighted in green as feedback. So the P's task is to select the word they heard, but given typical red-green colour associations, it seems they are told they are incorrect if they select the incongruent. So surely the opposite pattern of results was incredibly unlikely to materialise. Great if this point can be clarified and the conclusions altered accordingly.

2. The authors could explain better why specific techniques/models were used for specific subsets/hypothesis. At first glance it feels like they have a separate encoding/decoding/or both (back2back)/ rsm approach and then look either at beta weights, correlation coefficients, or variance explained. This gets quite convoluted and it would help to have more support for those choices. If these choices were preregistered this would help, but if not, why these different strategies?

3. As a specific, for the results in fig.2, what is the 'baseline' for each encoding improvement in 2D? Usually - and in the sense of a 'hierarchical' model - they would test baseline vs. 0, then acoustic vs. that baseline, and then semantic vs baseline + acoustic, which appears to be what they did according to model formulations in 2B (i.e. beta1 + beta2,3 + beta4,5). This however then rests on the assumption that the RDMs for acoustic and semantic invariant/specific are not highly correlated. If those RDMs look very similar, is this the most appropriate strategy? Clarification along these lines seems essential given the core claims about dissociations between acoustic and semantic predictions.

4. It was unclear what a speaker-general prior would give us. The extent to which you associate anything with anything else? The theoretical significance of this formulation should be clarified.

5. It's potentially very interesting if opposite effects are found according to acoustics vs semantics, but some further pointers would be helpful concerning how this resolves the conflicts in the literature. Are neural sharpening effects more common in sensory regions and prediction error effects in semantic processing locations? The authors consider how semantic error may be more useful for model updating but this presumably depends upon the level of the model. Interesting idea but it would seem to need further justification. In contrast the point about tasks (deviance detection) and the distinct neural effects they generate, was potentially very interesting. Maybe this latter point could receive more air time. Further justification for these claims given the current state of the literature would be very helpful.

Minor comments:

1. The manuscript contrasts 'predictive coding' with 'Bayesian' models. This is an unusual framing as predictive coding is often called a Bayesian brain theory. Predictive coding only emphasizes the processing of prediction errors in error units. In hypothesis units the opposite is true. I would therefore suggest replacing the name of the account specifying enhancement of an error signal as a predictive coding account. This could potentially lead to a misunderstanding of the theory, given the account only works because you have hypothesis and error units both playing key roles.

2. It would be nice to have more information concerning how the alternative word was chosen. It is claimed that the phonetic differences are minimal, but not how these relate to other contexts. Was each target paired equally with a word from each of the other five contexts?

Reviewer #2: In "Sensory sharpening and semantic prediction errors unify competing models of predictive processing in communication," Schneider and Blank address a central debate in the predictive coding literature: is speech comprehension facilitated by the sharpening of expected content (as proposed in the Bayesian brain framework), or by the minimization of prediction errors through enhanced encoding of unexpected input (as posited by hierarchical predictive coding, hPC)?

In a particularly elegant and methodologically rigorous study, the authors propose that these mechanisms are not mutually exclusive but can instead operate at different hierarchical levels of processing.

Using a well-designed behavioral paradigm, they first demonstrate that listeners can learn and apply speaker-specific semantic priors. Participants learned associations between six distinct speaker faces and six unique semantic contexts. During the task, a speaker's face cued the semantic context, followed by an acoustically morphed word blending two words, only one of which was congruent with the speaker's semantic context. The results show that listeners relied on speaker-specific priors to disambiguate the perceived word.

To probe the neural underpinnings of this effect, the authors employed representational similarity analysis (RSA) and similarity encoding models. They reconstructed gammatone-filtered spectrograms from EEG recordings and found that correlations between reconstructed and actual audio spectrograms were significantly above chance.

Next, they generated hypothesis RSMs based on the top-5 most probable acoustic predictions under each speaker-specific or -invariant prior and corresponding semantic predictions. They used encoding models to test which set of predictors best accounted for the similarity structure of the decoded EEG representations.

The results revealed that both speaker-invariant and speaker-specific priors shaped EEG encoding of acoustic input, with a stronger contribution from speaker-specific priors. To differentiate sharpening from prediction error mechanisms, the authors analyzed the β-coefficients of their best-fitting models. They showed positive coefficients for expected acoustic features under speaker-specific priors, indicating a sharpening effect: neural representations were biased toward predicted sensory input.

They then asked whether prediction error signals emerge at higher levels of the processing hierarchy. To address this, they constructed single-trial encoding models of broadband EEG data, incorporating acoustic and semantic surprisal measures derived from large language models as control predictors. Only speaker-specific semantic surprisal significantly improved model performance, suggesting that prediction errors arise only at high-level semantic stages and not from general phonotactic or lexical violations.

In a second EEG experiment, the authors tested the flexibility of semantic prior application by manipulating the congruency between prior expectations and the incoming speech signal. Participants heard vocoded but unmorphed words that were either highly congruent or incongruent with the speaker's semantic context. Behaviorally, incongruent trials resulted in longer reaction times, indicating a processing cost. Encoding models revealed a double dissociation: in incongruent trials, only speaker-invariant surprisal improved encoding performance, whereas in congruent trials, only speaker-specific surprisal did.

Together, this study demonstrates that sharpening and prediction error mechanisms coexist in speech comprehension: low-level sensory sharpening occurs early and is guided by semantic priors, while prediction error signals emerge at later stages.

I found the article very interesting, methodologically innovative, and well written.

The study makes a contribution to the predictive coding literature by elegantly demonstrating how sharpening and prediction error mechanisms coexist in the processing hierarchies.

I only have a few minor comments and suggestions for clarification:

- Figure 1D: The correlation coefficients reported are relatively low. Some contextualization or discussion would be helpful to interpret the strength of these effects.

The authors chose to rely on raw EEG data. This limits insight into the specific neurophysiological features (e.g., time-frequency dynamics, phase-amplitude coupling, evoked responses) that carry sharpening or PE information. Including additional analyses based on TF decomposition, Hilbert envelopes or phase amplitude coupling might help clarify the underlying mechanisms of sharpening or PE. While I understand that addressing this would require substantial new analyses, the authors could consider discussing these potential features in the discussion section.

- Finally, the spatiotemporally resolved contributions of speaker-specific semantic surprisal (Figure 3C) and speaker-invariant semantic surprisal in incongruent trials (Figure 4D, top), as well as speaker-specific surprisal in congruent trials (Figure 4D, bottom), appear across all channels. This provides limited insight into the underlying neural networks involved.

Given these points, I believe the manuscript would benefit from a discussion or, if feasible, supplementary source-level and oscillatory analyses addressing the electrophysiological mechanisms and brain network dynamics that may underlie these effects.

Reviewer #3: This manuscript presents research aimed at investigating the neural mechanisms underlying speech perception and how that perception is influenced by knowledge of an individual speaker's semantic priors. Specifically, the authors record EEG from participants as they listened to vocoded ambiguous (morphed) stimuli. Participants learned that certain speakers were more likely to speak about certain semantic topics and this influenced the perception of the ambiguous words. Using a combination of encoding/decoding modeling on the EEG and RSA the authors argue that the semantic priors sharpen the auditory representations of the ambiguous speech and lead to larger prediction errors at higher linguistic levels. They suggest that this leads to a more nuanced view of hierarchical predictive perception.

This was a well written manuscript that tackled an interesting question using a nice experimental design and sophisticated analyses. I thought the results were compelling and interesting and that the discussion was very reasonable. However, I do have a few comments - most of which center on the methods and the motivation for and description of those methods.

Main comments:

1) I must admit to finding it somewhat difficult to understand the motivation for the multistep analysis. For example, for the question of whether acoustic representations are sharpened or whether they reflect prediction error, why not simply directly compare (via correlation) the reconstructed spectrograms for the morphs heard in the context of different speakers with the actual spectrograms of the stimuli making up those morphs? Would the reconstructed spectrogram being more highly correlated with tea than sea in one context and more highly correlated with sea than tea in the other (at least relatively) not allow you to quite directly compare the sharpening and prediction error accounts? Why the need to push it all through the RSA framework? I am not saying what you have done is incorrect. Just that it might be worth more clearly explaining the advantages of the approach you have chosen.

2) Following on from the previous comment, one of the main reasons why I wondered about the motivation for the RSA framework is that I also wondered about the specific implementation. In particular, the notion of building hypothesis RSMs based on a weighted average of the spectrograms of different words seemed very strange to me. Two words could have very similar semantic meaning and very different spectrograms. Averaging five (or more) spectrograms together seems certain to smear those spectrograms quite heavily and make the interpretation of the RSMs quite opaque, no? Moreover, I failed to appreciate why some words were more probable for a speaker than any other words. Was this just based on word frequency within category? And what about for speaker-independent, just the five most frequent words over all categories? Again, I think the motivation for the two-stage analysis could have been better. It is complicated and involves some assumptions and approaches that need to be justified and better explained in my opinion.

3) Another question regarding the acoustic analysis: I struggled to understand from the methods section what was done for the speaker invariant analysis. I didn't see that explained in methods section 4.9 at all.

4) The method section on the linguistic surprisal analysis (4.10) was confusing too I am afraid to say. It starts off by arguing that one can't really just calculate phonotactic/lexical/semantic surprisal for morphed stimuli - hence the need to use activations from different layers of wav2vec. But then it describes how you computed phonotactic/lexical/semantic surprisal. Why? Based on both words in the morph? Was the idea to see how the activations from different wav2vec layers relate to these different kinds of surprisal? And then to use the activations - or the feature-based predictions of the activations (or vice versa) - in the EEG analysis? It's all a bit opaque. And then in figures 3 and 4 we are only told about semantic surprisal - not sure where phonotactic and lexical surprisal went. Is the idea that you wanted to identify how the wav2vec activations related to semantic surprisal while removing the influence of phonotactic and lexical surprisal? We are also told the best layer is chosen. The best layer with respect to what feature? Semantic surprisal? It's very unclear what was done.

5) BTW - is the speaker-specific vs speaker-independent semantic analysis also based somehow on the 5 most probable words?

Minor comments:

1) It might be worth very briefly describing your GLMM in section 2.1 of the main body. The results are very difficult to appreciate without knowing what the model was trying to ask.

2) Line 143 mentions "congruent words". Not sure what congruent means for this analysis/experiment.

3) What was the motivation for the stimulus creation step described in the paragraph beginning on line 508? Why the need to minimize the geometric means of Rt and Ra?

4) 20 words per context and a total of sixty morphs. So, each word is morphed with 3 others?

5) On lines 560-563… it seems you want find two words that are congruent and incongruent with a particular context. But you talk about a context_a and a context_b. Why?

Reviewer #4: The authors investigate how predictive coding and Bayesian models of speech comprehension might be reconciled. Using EEG with ambiguous speech presented in different speaker contexts, they test whether speaker-specific semantic priors sharpen sensory representations while prediction errors appear at higher linguistic levels. They propose that these mechanisms operate at distinct hierarchical stages of speech perception.

This is an ambitious paper, and some results - particularly regarding sharpening and the representational disambiguation of stimuli - are very interesting. However, because of methodological issues, I am not entirely convinced by the conclusions.

Before the details, I must note that the paper is written in an unclear and at times frustrating way. Many methodological choices are not well motivated, the use of technical terms often diverges from their standard usage, and explanations are unnecessarily dense, and cross-referencing is minimal. I spent much more time reviewing this paper than I normally would. The authors could do more to make it reader-friendly.

General comments:

1. RSA-like approach

The authors describe their analysis as RSM-based, citing Kriegeskorte's RSA work, but their use of RSMs differs substantially from standard practice. This introduces unnecessary complexity and potential statistical bias.

In typical RSA, RSMs are square matrices containing pairwise distances between items in a model space. Here, RSMs are rectangular matrices reflecting *within-item* distances between decoded, real, and hypothesized spectrograms. This design may lose valuable information about dissimilarities across items.

The terminology is also misleading. For example, the matrix containing cosine similarities between real and decoded spectrograms is called the "sensory RSM," implying a uniform sensory representation rather than a composite of two representational formats.

Finally, all acoustic RSMs include distances from the real spectrogram. As a result, regression models (~RSA analyses) measuring similarities between such RSMs may inflate scores because both are anchored to the same variable.

2. Prior-weighted averages

I am not convinced by the idea of prior-weighted spectrograms. It seems implausible that averaging the spectrograms of the top 5 (or 19) exemplars of a semantic category would yield a meaningful acoustic signal, given that the mapping between sound and meaning is arbitrary. I'd expect that predicted acoustics will average out. Such averaging makes more sense for semantic embeddings, where words in the same category cluster in vector space.

Also, how are spectrograms of different word lengths averaged? Could length differences alone drive the results? A long spectrogram could correlate better simply because the averaged template includes a long word.

For these reasons, I find cosine-distance analyses to these averaged spectrograms questionable.

3. Granularity of tests

The discrepancy between acoustic sharpening and acoustic prediction error findings might arise from different analysis granularities. Sharpening is assessed with representation-content measures, while acoustic prediction error is tested using a univariate cosine-distance metric.

4. Statistical analyses

I am concerned by the extremely low p-values (e.g., p < 10⁻¹⁹ for RTs). Such values often indicate mis-specified models, pseudo-replication, or overfitting.

Looking at the LMER model specification, they look quite unusual to me. E.g. the model from p. 28:

(1 + t + p(word|speaker)|participant × context) + (1|participant × position) + (1|participant × stimulus) + (1|face × feature)

is unconventional. The inclusion of random variables such as participants:stimulus indicates that there is a separate level for *each combination of item and participant*.

I presume 'x' in tje models means '\*' in lmer specification, i.e. full factorial design. I would expect that such models will not converge, and at the very least, it would be useful to remove random correlations.

I think the formulation should look more like:

lmer(Y ~ predictors +

+ (1 + applicable_predictors || participant)

+ (1 + applicable_predictors || context) # I was unsure what context refers to

+ (1 + applicable_predictors || stimulus)

Minor comments

1) p.6 - "Firstly, we built acoustic hypothesis RSMs from spectrograms of clear words that were predictable from participants' speaker-specific and -invariant semantic priors."

- How can you know which words were expected to participants when discarding information about the speaker?

- How do you know which category-specific words were the most predictable knowing the speaker? There are probably dozens of exemplars fitting a semantic category. Also, before the experiment, participants don't know which 20 exemplars will be used.

2) p.6 - "Secondly, we generated semantic hypothesis RSMs by comparing the semantic embeddings of the congruent words with the current estimates of the speaker-specific and speaker-invariant semantic priors (see Fig. 2b)"

- What were the congruent words? I assume you are not referring here to the next procedure in which you used congruent and incongruent words, but rather mean the speaker-congruent part of the morphed word?

- I assume speaker-specific priors are computed using the free-energy optimization? if yes, an explicit link to that section of the paper would be useful

3) p. 6 - "Encoding models" is misleading; in neuroscience, this usually means predicting brain activity from stimuli, not regressing RSMs on each other. It would be clearer of the TRF analysis was called TRF.

4) p. 11 - Since you mention EMTs, at what reference values they were computed?

5) p.23 - What was the control predictor? Please explain its purpose, what were the sampled priors?

6) p. 23 and elsewhere - p(speaker) implies the probability of a specific speaker. But after careful reading one realizes that the authors mean instead a semantic prior distribution from the free-energy model. Possibly names of other variables in formula could also be clarified.

7) p. 26-27 - The "semantic prediction error" control variable (cosine distance from GloVe embedding to mean embedding of all words) seems ill-motivated. If the lexicon's centroid is near zero, the angle is unstable and not interpretable.

8) p.26 - "Note that coefficients w(t, n) in these models are not strictly temporal response functions given that we used standardised (i.e., dense) design matrices which allowed us to model non-linearly additive effects of consecutive events, at the cost of interpreting coefficients in reference to baseline levels rather than as discrete events." - It is unclear to me what the authors mean by "standardised (i.e. dense) design matrices", and how it implies nonlinearity. If the continuous nature of wav2vec is a problem, you could always average wav2vec embeddings within the analysed unit (phonemes or words) to convert them to discrete events, i.e. spike regressors. But maybe I misunderstood the problem.

9) p. 27 - Could you report how much variance was explained by the baseline models and how it improved by adding acoustic and semantic surprisal measures?

10) p. 27 - Statistical inference - what exactly was compared in the t-test (and in other places where models are compared)?

11) p.27 - "Baseline models also included either projected LLM activations or direct measures of phonotactic, lexical and semantic surprisal, relative to the target word" - I'm confused by the word "or".

12) I think the authors could cite Broderick et al., 2019 - who also showed results consistent with acoustic sharpening in auditory speech comprehension.

---

## [Decision Letter · Decision Letter 2]

4 Dec 2025

Dear Dr Schneider,

Thank you for your patience while we considered your revised manuscript "Sensory sharpening and semantic prediction errors unify competing models of predictive processing in communication" for publication as a Research Article at PLOS Biology. This revised version of your manuscript has been evaluated by the PLOS Biology editors, the Academic Editor and the original reviewers.

Based on the reviews and on our Academic Editor's assessment of your revision, we are likely to accept this manuscript for publication, provided you satisfactorily address the remaining points raised by the reviewers. Please also make sure to address the following data and other policy-related requests:

* We would like to suggest a different title to improve its accessibility for our broad audience: "Sensory sharpening and semantic prediction errors unify competing models of predictive processing in human speech comprehension"

* Please include the approval/license number of the ethical approval for the experiments.

* Please include information in the Methods section whether the study has been conducted according to the principles expressed in the Declaration of Helsinki.

* DATA POLICY:

Regardless of the method selected, please ensure that you provide the individual numerical values that underlie the summary data displayed in the following figure panels as they are essential for readers to assess your analysis and to reproduce it: 2CD, 3B, 4BC, S2, S4 (left panel), S6, S7, S8, S10 (left panel), S11 (left panel), S12 and S13C.

* CODE POLICY

* Please move the references from the the supplementary information into the main reference list, because the reference databased do not pick up citations from the supplementary information, so the authors won't be credited for these otherwise.

* Please move the methodological details from the supplementary information into the main methods section. We do not have a word count limit and want to make it as easy as possible to access all methodological information.

We expect to receive your revised manuscript within two weeks.

*Published Peer Review History*

*Press*

Sincerely,

Christian

Christian Schnell, PhD

Senior Editor

cschnell@plos.org

PLOS Biology

Reviewer remarks:

Reviewer #1: The authors have satisfactorily addressed our concerns and we think it will make an interesting contribution to the literature.

Reviewer #3: I thank the reviewers for their extensive efforts in responding to my previous comments (and those of the other reviewers). I found most of the responses very satisfactory and think the manuscript is much improved overall. However, I have to admit that I was not satisfied by the response to my first comment from the previous round of review. Specifically, in my reading, the manuscript still lacks a motivation for the use of the RSA approach. Again, I am not saying there is anything wrong with the approach, but no motivation for using it is provided. The authors say in their response that they "explain why simply comparing morph sea-tea to the constituent words is inadequate". But I didn't see that explanation anywhere. Why is it inadequate? And why is the RSA not necessary when it comes to carrying out the analysis on semantic priors? For that analysis the RSA stage seems to be jettisoned and the analysis focuses on comparing the ability of different models to explain EEG data directly. This begs the question as to why one cannot do something similar for the sensory responses and just compare acoustic reconstructions directly? So again, the use of the RSA needs to be motivated with some text early in the main body of the manuscript so that readers can appreciate its importance. Otherwise, readers are likely to be unnecessarily confused or skeptical of the strength of the results or both.

---

## [Editor Report · Decision Letter 3]

17 Dec 2025

Dear Dr Schneider,

Thank you for the submission of your revised Research Article "Sensory sharpening and semantic prediction errors unify competing models of prediction in human speech comprehension" for publication in PLOS Biology. On behalf of my colleagues and the Academic Editor, Manuel Malmierca, I am pleased to say that we can in principle accept your manuscript for publication, provided you address any remaining formatting and reporting issues. These will be detailed in an email you should receive within 2-3 business days from our colleagues in the journal operations team; no action is required from you until then. Please note that we will not be able to formally accept your manuscript and schedule it for publication until you have completed any requested changes.

PRESS

Sincerely, 

Christian

Christian Schnell, PhD

Senior Editor

PLOS Biology

cschnell@plos.org